# Non-Programmers Can Label Programs Indirectly via Active Examples: A Case Study with Text-to-SQL

**Ruiqi Zhong**[*][1]      **Charlie Snell**[*][1]      **Dan Klein**[1]      **Jason Eisner**[2]

[1] University of California, Berkeley [2] Microsoft Semantic Machines
{ruiqi-zhong, csnell22, klein}@berkeley.edu
jason.eisner@microsoft.com

## Abstract

Can non-programmers annotate natural language utterances with complex programs that represent their meaning? We introduce APEL, a framework in which non-programmers select among candidate programs generated by a seed semantic parser (e.g., Codex). Since they cannot understand the candidate programs, we ask them to select indirectly by examining the programs' input-ouput examples. For each utterance, APEL actively searches for a simple input on which the candidate programs tend to produce different outputs. It then asks the non-programmers only to choose the appropriate output, thus allowing us to infer which program is correct and could be used to fine-tune the parser. As a case study, we recruited human non-programmers to use APEL to re-annotate SPIDER, a text-to-SQL dataset. Our approach achieved the same annotation accuracy as the original expert annotators (75%) and exposed many subtle errors in the original annotations.

## 1 Introduction

Semantic parsing often aims to map a natural language utterance to a program, which can be executed on an input (Kushman and Barzilay, 2013). For example, for the utterance "*How old is the youngest person,*" we can map to the SQL program SELECT MIN(AGE) FROM PEOPLE, execute it on an input database, and return the output answer. Language models like Codex often achieve substantial few-shot performance (Chen et al., 2021a). However, user-facing applications often involve novel phrases or domain-specific values unseen during pre-training or generic instruction-tuning, so the model still needs more labels for training. Since hiring programming experts to label training data is costly, can we ask non-programmers to label them?

We introduce **APEL**, a framework that indirectly labels programs with non-programmer responses on carefully constructed input-output examples. APEL must be seeded with a moderately good semantic parser: we use Codex for this in our experi-

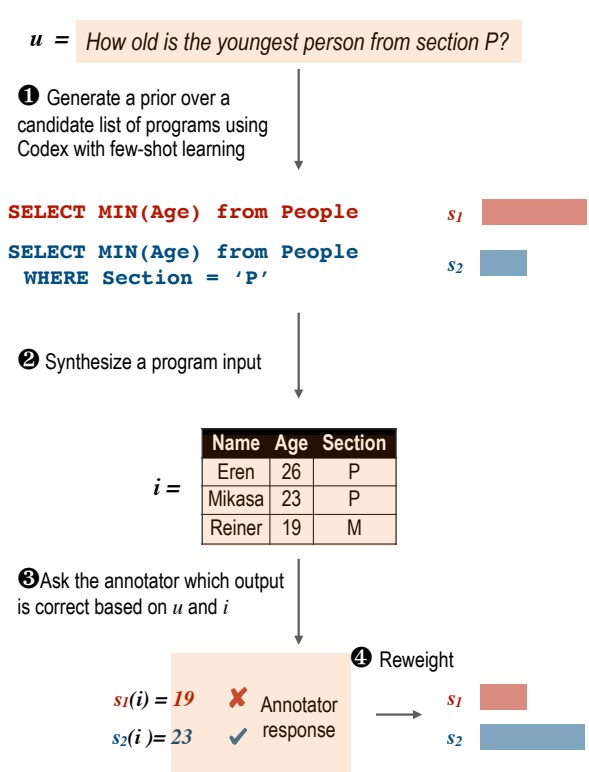

Figure 1: The outline of APEL. The orange background indicates what the annotators can see : they only need to select which output (19 or 23) is correct by reading the question $u$ and the database $i$. We downweight program candidates that are inconsistent with the annotators' response. Figure 3 illustrates our criteria for $i$.

ments, prompting it with a few demonstrations that we solicited from expert programmers. We now want to label additional training data. To construct the correct program for a new utterance, APEL uses the seed parser to generate a list of candidate programs with prior probabilities (Figure 1 top), and then asks non-programmer annotators to indicate which candidate is correct.

However, we cannot ask the annotators to select the correct program directly, since even expert programmers might find this challenging (Figure 2). Instead, APEL obtains information indirectly by asking questions about how the program should *be-*

Utterance u: Find the first name of students who have both cat and dog pets.

```
        SELECT t1.fname
        FROM student AS t1
        JOIN has_pet AS t2 ON t1.stuid = t2.stuid
        JOIN pets AS t3 ON t3.petid = t2.petid
s₁=     WHERE t3.pettype = 'cat' INTERSECT
        SELECT t1.fname
        FROM student AS t1
        JOIN has_pet AS t2 ON t1.stuid = t2.stuid
        JOIN pets AS t3 ON t3.petid = t2.petid WHERE t3.pettype = 'dog'

        SELECT fname
        FROM Student
        WHERE StuID IN
          (SELECT T1.stuid
           FROM student AS T1
           JOIN has_pet AS T2 ON T1.stuid = T2.stuid
s₂=        JOIN pets AS T3 ON T3.petid = T2.petid
           WHERE T3.pettype = 'cat' INTERSECT
           SELECT T1.stuid
           FROM student AS T1
           JOIN has_pet AS T2 ON T1.stuid = T2.stuid
           JOIN pets AS T3 ON T3.petid = T2.petid WHERE T3.pettype = 'dog')
```

Figure 2: APEL needs to obtain information about whether $s_1$ or $s_2$ is correct; it is hard even for experts to directly select the correct one by eyeballing. The error of $s_1$ is subtle: it finds the first names of the students who have dogs and the first names of the students who have cats, and intersects these two sets of first names. APEL can reveal this error by asking about a tiny student database $i$ in which Alice Jones owns a dog while Alice Smith owns a cat. As no student named Alice owns both, the annotator will select an output $o$ that does not include Alice, implying that $s_1$ is incorrect.

*have*. We synthesize a program input, execute the candidate programs on it to obtain a list of outputs, and ask the annotators which output is correct given the utterance and the input. We could *eliminate* the programs that returned other outputs, but annotators may make mistakes, so we instead *downweight* them via Bayes' Theorem (Figure 1 bottom). To pin down the correct program, we iteratively reduce the entropy of the distribution over candidates by repeating the process with more synthesized inputs.

To reduce annotators' workload, we propose an algorithm to search for a program input that maximizes the expected information gain of the annotator's answer (§3), subject to the constraint that the input is simple enough for the annotator to reason about the correct output easily (Figure 3). Since our method resembles the programming by example (PBE) framework (Lavrac and Dzeroski, 1994) and learns an underlying program by actively choosing which examples should be labeled, we call our framework APEL—**A**ctive **P**rogramming by **E**xamples using a prior derived from a natural **L**anguage utterance.

**Non-Programmer Annotator Study.** As a case study, we use APEL to label SQL programs. We built an annotation GUI based on APEL and recruited 11 non-programmers to label a random subset of 240 utterances from the SPIDER (Yu et al.,

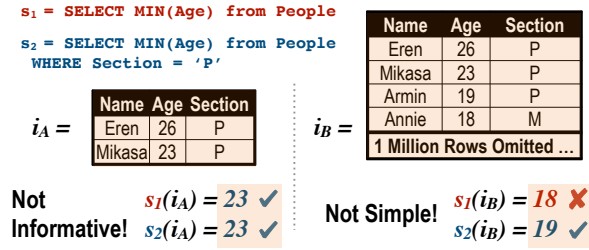

Figure 3: Informativity and simplicity guide our search for a useful input. $i_A$ is not informative since knowing the correct output does not tell us which program $s$ is correct. $i_B$ is not simple since the input is too large for humans to find the correct output. Figure 1 shows an ideal input.

2018) development set (§6). According to a new carefully constructed gold standard, we achieve the same accuracy as the original SPIDER annotation performed by database experts (75%) and also substantially outperforms the top-1 accuracy of Codex (59%). APEL also exposes subtle errors made by previous experts, which we analyze in §6.6.

APEL is a preliminary step towards enabling non-programmers to label utterances with arbitrarily complex programs. Applying it to new semantic parsing tasks still faces some bottlenecks: it requires a seed semantic parser with reasonable top-$k$ accuracy and an efficient procedure to find simple and informative program inputs (§7). However, these bottlenecks might be alleviated by future advances in semantic parsing and program synthesis. With these advances, we hope APEL can facilitate non-programmers to label programs in a broader range of applications in the future.[1]

## 2 Framework

APEL aims to enable humans to indirectly annotate natural language utterances with programs. Here we use text-to-SQL as a case study.

### 2.1 Outline of APEL

Let $u$ be a given utterance and $c$ a known database schema, which specifies the table names, column names, and the constraints on value types, uniqueness, and foreign keys. We want to synthesize a SQL program $s$ that captures the meaning of $u$ and works properly for *any* database with schema $c$.

Figure 1 illustrates our pipeline. We first feed $c$ and $u$ to a seed semantic parser (e.g. Codex with

---

[1]The corrected annotations and our code for searching for informative and simple databases are on our github: `https://github.com/ruiqi-zhong/EMNLP23-APEL` .

a prompt) to generate a prior distribution $p$ over a list of SQL programs $s$. We then 1) synthesize an input database $i$, 2) execute the list of programs on $i$ to obtain a list of program outputs $o$, and 3) display $u$, $i$, and the outputs to an annotator $a$. The annotator generates a response $r$ indicating which output they think is correct. We will then raise the posterior probabilities of the candidates $s$ such that $s(i) = r$, i.e. $s$ returns $r$ when executing on $i$. Since we ask the annotator to select the correct output of a program given an input, we call our questions "*o-selection questions.*"

To formalize, the posterior distribution over the program $s$, once we observe the response $r$, is

$$p(s \mid u, a, i, r) \propto p(s \mid u, a, i) \, p(r \mid s, u, a, i)$$
$$= p(s \mid u) \, p(r \mid a, s(i)) \quad (1)$$

where $p(s \mid u)$ was the prior from the seed parser. Here $p(r \mid a, s(i))$ models annotator behavior: the probability that annotator $a$ would have responded with $r$ if $s$ were a correct implementation of $u$ and therefore $s(i)$ were the correct output. To the extent that annotators are accurate, this Bayesian update increases the probability of the correct program.

We can ask more $o$-selection questions to obtain further information about the correct candidate and improve the posterior. If desired, different rounds may use different annotators. For each $u$, we define

$$p_t(s) \stackrel{\text{def}}{=} p(s \mid u, a_1, i_1, r_1, \dots, a_t, i_t, r_t)$$
$$\propto p_{t-1}(s) \, p(r_t \mid a_t, s(i_t)) \quad (2)$$

to be the posterior after $t > 0$ questions, with $p_0 \stackrel{\text{def}}{=} p(s \mid u)$ being the prior. Letting $T$ denote our total number of questions about $u$, our final posterior is $p_T$.

**Evaluation.** We output as our annotation the most probable SQL candidate $\hat{s}_T$ according to $p_T$, and compare it to a gold standard. To show that our framework is useful, $\hat{s}_T$ needs to be correct more often than $\hat{s}_0$, the most probable SQL candidate according to the seed semantic parser $p(s \mid u)$.

**Relation to Weakly Supervised Learning.** Appendix A explains how the "soft annotations" provided by the full distribution $p_T$ could be used to retrain the semantic parser $p(s \mid u)$. These improved models would in turn improve the estimate of $p_T$ (i.e., an EM algorithm).

## 2.2 Criteria for Synthesized Inputs

To generate an $o$-selection question on round $t$, APEL needs to synthesize an input database $i_t$ that

is both **informative** and **simple**.

**Informative.** Our belief as we enter round $t$ is $p_{t-1}$. Once we observe the annotator's response $r_t$, we will be able to update it to $p_t$. This will achieve an information gain of $\mathrm{H}(p_{t-1}) - \mathrm{H}(p_t)$, where the Shannon entropy $\mathrm{H}$ of a distribution over programs $s$ characterizes its remaining uncertainty about which program is correct.

However, $p_t$ will depend not only on our choice of question $i_t$ but also on the annotator's response $r_t$ (equation (2)). We do not know $r_t$ yet, but our current belief is that it will be distributed as

$$p_{t-1}(r_t) = \sum_s p_{t-1}(s) \, p(r_t \mid a_t, s(i_t)) \quad (3)$$

So our *expected* **I**nformation **G**ain from asking $i_t$ is

$$\mathrm{IG}_{p_{t-1}}(i_t) \stackrel{\text{def}}{=} \mathrm{H}(p_{t-1}) - \mathbb{E}_{r_t \sim p_{t-1}}[\mathrm{H}(p_t)] \quad (4)$$

where the expectation is taken under distribution (3). This is high if the candidates $s$ that are most plausible under $p_{t-1}$ tend to return different outputs $s(i_t)$ and hence different annotator responses $r_t$, making $r_t$ informative about $s$. By contrast, $i_A$ in Figure 3 left would yield an uninformative response ($\mathrm{IG} = 0$).

**Simple.** We would like to avoid presenting the annotator with complex inputs such as the large database $i_B$ in Figure 3 right. The correct response might be informative, but determining it would require too much human effort. We crudely model the effort required for $i_t$ as the number of records $|i_t|$.

The next section proposes a heuristic to synthesize a simple informative database $i_t$ given schema $c$, a sample database with schema $c$, and a distribution $p_{t-1}$ over SQL programs.

## 3 Optimizing the Input Database

We attempt to maximize the expected information gain IG over all databases that conform to the given schema $c$ and have at most $R$ total records, where $R = 30$ in our case study. Formally, we search for

$$i_t^* = \underset{i_t : |i_t| \le R}{\mathrm{argmax}} \, \mathrm{IG}_{p_{t-1}}(i_t). \quad (5)$$

When multiple databases have the same IG, we break ties by favoring the smallest database. Since $t$ and $p$ are fixed during the optimization process, we will write $\mathrm{IG}(i)$ instead of $\mathrm{IG}_{p_{t-1}}(i_t)$ for short.

Our method can be summarized as "fuzz-then-drop." Fuzzing (Miller et al., 1990) is an established practice in software testing, where an algorithm generates a large number of random program

inputs to search for an input that satisfies a property or reveals a bug. In our case, we want to search for an input database that maximizes the information gain. Therefore, we first perform fuzzing by randomly generating a large number of large databases as in Zhong et al. (2020)—see Appendix C for further details—and keep the database $i^0$ that maximizes the expected information gain $\text{IG}(i^0)$. We then drop records from $i^0$ to satisfy the simplicity criterion for $L$ iterations.

For each iteration $\ell$ of dropping records, we randomly drop 5% of the records from $i^\ell$ to obtain $i^{\ell+1}$. If this results in a less informative database, i.e., $\text{IG}(i^{\ell+1}) < \text{IG}(i^\ell)$, we are willing to retry up to 20 times in hopes of randomly finding an $i^{\ell+1}$ that is at least as informative as $i^\ell$; if we fail in all 20 tries, we keep the best of the 20 despite the decrease in IG. Of the databases smaller than $R$ that we encountered during the $L$ iterations, let $\hat{i}$ be the one with the highest IG:

$$\hat{i} \overset{\text{def}}{=} \underset{i \in \{i^\ell : 1 \le \ell \le L\}, |i| \le R}{\text{argmax}} \text{IG}(i) \qquad (6)$$

Since our procedure is randomized, we repeat it 3 times, and let $i^*$ be the $\hat{i}$ with the largest $\text{IG}(\hat{i})$, breaking ties as before in favor of smaller $\hat{i}$. Finally, we simplify $i^*$ by dropping tables and columns that were not mentioned by any SQL candidates in $p$.

Our procedure of dropping records from a large informative database is heavily inspired by Miao et al. (2019), which, given a database $i$ such that $s_1(i) \ne s_2(i)$, provably finds the smallest subset of records in $i$ such that programs $s_1$ and $s_2$ return different outputs. However, their algorithm works only for a restricted family of SQL programs and cannot be adapted to optimize information gain. Our procedure does not provide any provable optimality guarantee, but is more flexible and practical.

In practice, simply applying the above procedure can generate unnatural databases and lead to vacuous SQL execution, confusing the annotators. In Appendix C, we illustrate several typical confusions (Figure 6) and explain how we fix them.

## 4 Experimental Setup

We describe the dataset used to benchmark APEL (§4.1), how we obtained the prior over SQL candidates (§4.2), and our evaluation metrics (§4.4).

### 4.1 Dataset

We benchmarked APEL on the development set of SPIDER (Yu et al., 2018), an English text-to-

SQL dataset with 1034 utterance-SQL pairs distributed under the CC BY-SA 4.0 License. The development set is divided into 20 *domains*, each with a distinct database schema $c$, a collection of utterance-SQL $(u, s)$ pairs, and a sample database. We split the 1034 $(u, s)$ pairs equally into a *validation split* and an *evaluation split*. We used the validation split to tune our annotator interface (§6), develop our fuzz-then-drop algorithm (§3), and prompt Codex (§4.2) to create a seed parser. We used the evaluation split to evaluate APEL with simulated annotators (§5), and from these drew a random subset of 240 utterances balanced across domains to conduct our human evaluation (§6).

To make our evaluation more reliable, we 1) corrected the sample database to conform to the schema $c$ and updated the test suite correspondingly (Appendix D), and 2) established a new gold standard by correcting errors in 61 out of these 240 SQL annotations (§6.2). The corresponding author of SPIDER endorses our corrections (T. Yu, p.c.).

### 4.2 Obtaining the SQL Program Prior $p_0$

We generated a prior over SQL program candidates using the Codex (Chen et al., 2021b) language model with few-shot prompting. Given an utterance $u$ with a schema $c$, we created the prompt (Appendix E.1) by concatenating a linearization of $c$ with eight distinct $(u_k, s_k)$ pairs from the validation split associated with the same schema $c$, and finally the utterance $u$ itself. Some $(u_k, s_k)$ pairs were chosen randomly while others were chosen because $u_k$ has the highest TF-IDF similarity to $u$. We randomly sampled 200 prompts for $u$ by choosing different $(u_k, s_k)$ pairs, and for each prompt, we asked Codex to generate 20 completions (SQL programs) and filtered out non-executable candidates. To prevent surface form competition (Holtzman et al., 2021), we then merged approximately semantically equivalent candidates by finding sets of candidates that return exactly the same outputs on 1K random databases, using the implementation from Zhong et al. (2020). We define $p_0$ to be the empirical distribution in our sample of the 16 semantic equivalence classes that were most frequent in the sample. Thus, each $s$ in §2 represents an approximate equivalence class rather than a single program.

Treating the original SPIDER annotation as the ground truth, the top-1 accuracy of $p_0$ on the entire development set is 72% and the top-16 accuracy is 94%. These numbers are not comparable

to prior works, which usually evaluate on unseen database domains in a zero-shot manner (harder than our setting) but do not predict string literals and `DISTINCT` keywords, which we need for execution. Appendix E.2 includes more details on top-$k$ accuracy (sometimes also referred to as pass@$k$).

Note that the seed parser could be further improved, e.g. with better models (OpenAI, 2023) or prompts (Zhou et al., 2023). However, these improvements are complementary to APEL, whose goal is to improve over the seed parser with non-programmer responses. We design our evaluation metrics based on this goal in the next section.

### 4.3 Separation of Annotators

When optimizing the next input $i$ to show to a given annotator $a$ (§3), we measured its IG by conditioning on the previous responses from $a$ only. That is, we treated each annotator as if they were the first annotator. (Indeed, all annotators for $u$ saw the same first question.) This approach reduced the amount of information about $u$ that our $T$ total questions could extract, but it also reduced the variance of our experiment by running multiple *independent* (but shorter) annotation sessions on $u$. To compute the final $p_T$ for $u$, we still aggregated the responses from all annotators using the Bayesian formula (2).

### 4.4 Evaluation Metrics

We report the following statistics to evaluate APEL:
- *Codex accuracy*: how often is the argmax of $p_0$ correct? This baseline reflects the top-1 accuracy of the seed semantic parser.
- *APEL accuracy*: how often is $\hat{s}_T$ (the argmax of $p_T$) correct? This reflects the improvement due to our APEL annotators.
- *candidate ceiling*: how often does the support of $p_0$ contain a correct program? This reflects how much APEL is bottlenecked by the seed semantic parser, whose top 16 candidates form our candidate list.

If APEL is effective, we should see that the APEL accuracy is higher than the Codex accuracy. Since SPIDER categorizes its utterances by difficulty (easy/medium/hard/extra), we report the accuracy for each difficulty level. Next we evaluate APEL with both simulated and human annotators.

## 5 Automated Simulation Evaluation

To automatically test the effectiveness of APEL without additional human annotations, we bench-

|          | easy | med  | hard | extra | all  |
|----------|------|------|------|-------|------|
| Ceiling  | 0.98 | 0.96 | 0.98 | 0.81  | 0.94 |
| Codex    | 0.87 | 0.80 | 0.56 | 0.45  | 0.72 |
| OrigDB   | **0.98** | 0.90 | 0.83 | 0.66  | 0.86 |
| APEL     | 0.93 | **0.95** | **0.97** | **0.75** | **0.91** |

Table 1: The description of the metrics can be seen in §4.4. APEL means using databases generated using the method described in §3 while OrigDB directly uses the sample database from SPIDER. APEL significantly outperforms both Codex and OrigDB with $p$-value $< 1 \times 10^{-3}$, indicating that both APEL and our database generation algorithm are effective.

marked APEL on the entire evaluation split by simulating an oracle annotator who always responds correctly by choosing the output of the correct SQL program and assuming that the SQL annotation provided by SPIDER is always correct. Additionally, to evaluate our database generation algorithm in §3, we compared to OrigDB, an ablated variant of APEL that directly uses the sample database from the original SPIDER dataset. Table 1 reports the candidate ceiling, the Codex accuracy, and the accuracy for both OrigDB and APEL.

With 3 rounds of interactions, APEL accuracy (91%) substantially improves on the Codex accuracy (72%), validating the effectiveness of our framework; we achieved this by interacting with our oracle annotator for 1.8 rounds on average. Compared to OrigDB (86%), which uses the sample databases from the original SPIDER dataset to interact for 1 round, APEL's database generation method leads to higher accuracy since it allows multiple rounds and optimizes for information gain. Furthermore, averaged across all utterances, the databases generated by APEL contained altogether 10 records across all rounds—whereas the OrigDB databases contained on average 33,295 records and would be impractical to present to human annotators. These results highlight the need for the databases to be both simple and informative.[2]

Appendix H provides further information on the database sizes, the number of interaction rounds, and the robustness of our database generation method under different hyper-parameter choices.

## 6 Human Evaluation

We evaluate the effectiveness of APEL with real human annotators. We built an interface designed

---

[2]APEL achieves 81% accuracy with 1 round of interaction.

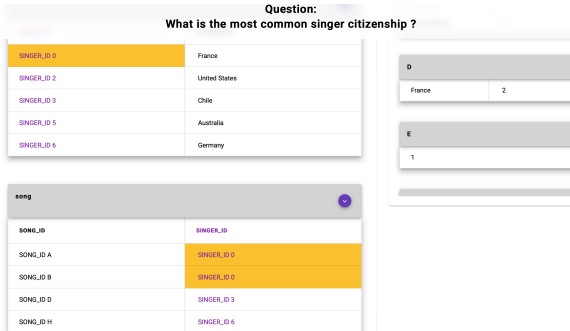

Figure 4: A screenshot of our annotation interface (§6.1). Appendix J and Figure 11 include more details.

to be user-friendly (§6.1) and used it ourselves to establish gold annotations for 240 utterances (§6.2). We then recruited 11 non-programmer subjects to annotate the same utterances (§6.3), aggregated their responses by learning a model of annotator behavior (§6.4), and benchmarked their performance with the newly established gold standard (§6.5). We analyze errors by SPIDER expert annotators in §6.6, qualitative feedback from our subjects in §6.7, and errors made by our subjects in Appendix F.

## 6.1 Annotation Interface

Figure 4 shows an example $o$-selection question in our interface, which displays the utterance $u$ on the top, the database $i$ on the left, and up to $M = 6$ distinct outputs $o$ on the right in random order, followed by a "none of the above" option; generally, an output may be a string, a number, or a table. The annotator can also use an open-ended response field to optionally report that the question is ambiguous or confusing; we did not use these open-ended responses during data collection, but future work could potentially condition on them in equation (2) (along with the multiple-choice responses).

To reduce the annotators' cognitive load, when the mouse pointer is over a cell, our interface highlights that cell along with all other cells of $i$ and of the outputs $o$ that have the same value. Appendix J describes more features of our interface.

We asked each annotator up to 3 consecutive questions for each utterance $u$, stopping the interaction after 1 or 2 questions if some candidate $s$ already has $p_t(s) > 0.9$, or if our heuristic (§3) fails to find an appropriate database $i$ for the next question.

## 6.2 Gold Standard Annotation

To establish a clean gold standard, two of the authors annotated all 240 utterances using our own APEL system. Whenever our two responses to a APEL question were different, we reached a consensus through discussion.

We closely examined the $o$-selection questions where our consensus response did not match the output of the SQL program provided by SPIDER, and corrected SPIDER's program if we felt that our response was strictly better. To avoid biasing against the original annotations, we stuck to the original ones whenever there were ambiguities, and we double-checked each corrected annotation by additionally writing down reasons why we felt it was better. As mentioned in §4.1, we ultimately corrected 61 out of the 240 SQL annotations. §6.6 analyzes these corrections in greater detail.

## 6.3 Non-Programmer Annotation

**Annotation Unit.** We split the 240 utterances into 8 units. Each unit contained 30 utterances across 4–5 database domains and proved to take 1–2 hours to annotate with our interface. For the annotator behavior model $p(r \mid a, s(i))$ in equation (1), we assumed that every annotator would respond correctly with 0.7 probability and would otherwise select a response uniformly at random.

**Recruiting Non-Programmers.** As annotators, we recruited 11 university students who 1) were not pursuing/had not received a Computer Science degree and 2) had no prior experience with SQL. Each annotator could annotate any number of units (from 1 to 8) as they wished, but had to annotate them fully. For each unit we rewarded them with $15 as a base payment and a $5 ($10) bonus if their response agreed with our corrected gold standard $> 85\%$ (95%) of the time. We received 20 units of annotation in total, and hence each utterance was examined by 2.5 annotators on average. We asked each of them 1.84 questions about the utterance and the databases that we presented contain only 8.05 records on average. We visualize the distribution of the number of questions for each utterance and the database sizes for each question in Figure 5.

**Participation Procedure.** We asked each annotator to: 1) sign a consent form to participate in the study, 2) watch a 12-minute video tutorial that contains our annotation instructions and explains the basics of foreign and primary keys, and 3) complete the annotation task. The tutorial can be

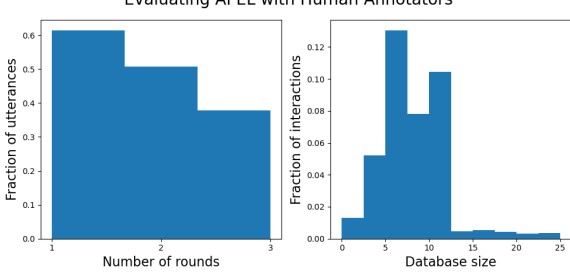

Figure 5: The distribution of database sizes (left) and number of questions (right) for the human evaluation.

|  | easy | med | hard | extra | all |
|---|---|---|---|---|---|
| Ceiling | 0.91 | 0.91 | 0.92 | 0.69 | 0.88 |
| SPIDER | 0.93 | 0.78 | 0.63 | 0.50 | 0.75 |
| Codex | 0.78 | 0.65 | 0.43 | 0.36 | 0.59 |
| APEL | **0.83** | **0.80** | **0.73** | **0.47** | **0.75** |

Table 2: Similar to Table 1, except that we now use the gold standard from §6.2 and also report the annotation accuracy of the original SPIDER dataset. APEL significantly outperforms Codex ($p$-value $< 10^{-3}$).

seen at https://youtu.be/-MlIcCQ21xs and an example unit of the annotation task can be tried at http://35.225.126.31:4200/v0104_4pm_8.

### 6.4 Learning an Annotator Behavior Model

After collecting the annotations, we used them to improve §6.3's naive model of annotator behavior $p(r \mid a, s(i))$ by learning parameters $\alpha_a$ for each annotator $a$ and $\beta_d$ for each SPIDER domain $d$. For a given $a$ and $d$, the chance that the annotator answers at random is no longer fixed at 0.3, but is modeled as $\sigma(\alpha_a + \beta_d + b)$, where $\sigma$ is the logistic function and $b$ is a bias term. Larger $\alpha_a$ and $\beta_d$ predict higher error rates.

We maximize the incomplete data log-likelihood

$$L_u = \log p(r_1, \ldots r_T \mid u, i_1, \ldots i_T)$$
$$= \log \sum_s p_0(s) \prod_{t=1}^T p(r_t \mid a_t, s(i_t)) \quad (7)$$

summed over all utterances $u$. Since we explicitly model the annotator behavior, $L_u$ is sensitive to the domain $d$ of $u$ and the annotator $a_t$ who answered the question about $i_t$. Just as in other adaptive crowdsourcing research, we do not assume access to the gold value of $s$. Our behavior model will predict a lower annotation error rate for those who tend to agree with other annotators and with Codex.

Overall, our annotators chose the correct option 72% of the time. To evaluate our unsupervised annotator model, we compared its predicted probability of a correct response on each utterance to the 0/1 indicator of whether the annotator did select the correct option. See Appendix I for a visualization. Our model achieves an AUC ROC score of 0.68 and MSE error of 0.185. For comparison, a simple *supervised* model that always predicted the true overall probability of 72% would achieve an MSE error of 0.20.

As Appendix A explains, one way to maximize equation (7) is to use an EM algorithm that alternates between imputing the unknown $s$ for each utterance (using $p_T$) and re-estimating the annotator error model based on these imputed "gold" answers. Appendix A also discusses how the annotator error model could be enriched.

### 6.5 Results

We present the results in Table 2. APEL (75%) is effective, since it significantly outperforms Codex (59%) with $p$-value $< 10^{-3}$ under a one-sided paired t-test. APEL leads to the highest improvement for utterances for medium or hard difficulty levels, implying that APEL is most effective when the utterances are hard enough so that there is still room for improvement, but not too hard to confuse the non-programmer annotators. Additionally, APEL with non-programmers is on par with previous database experts without the help of a seed parser. We next analyze the errors made by previous database experts.

### 6.6 Subtle Errors by Database Experts

APEL helped us identify annotation mistakes in the original SPIDER dataset (§6.2). We categorize and present them below to demonstrate where APEL is most helpful. The corresponding author of SPIDER agrees with our analyses (T. Yu, p.c.).

**Interpreting Database Schema Properly.** Suppose each row contains information about an orchestra, including the year the orchestra was founded and its associated recording company. For an utterance "*Which recording company was founded the earliest?*", the correct annotation under this schema should be "Not enough information to tell". Neither SPIDER nor APEL currently supports this annotation option, though our human annotator reported this issue in their open-ended feedback. SPIDER annotates this utterance incorrectly as SELECT company from TABLE WHERE YEAR =

(`SELECT MIN(YEAR) from TABLE`), which looks plausibly correct but actually finds the recording company of the earliest-founded orchestra.

**Handling All Allowed Values.** The annotated SQL should behave appropriately on all plausible cell values. For example, when we are asked about the maximum value in a column that allows `NULL` cells, we prefer a SQL that skips any `NULL` cells and returns the maximum of the actual values. As another example, if the utterance is "*How many countries have a republic government form?*", the clause `WHERE GOVERNMENT = "Republic"` will ignore any countries with the government form "*Federal Republic*"; the correct annotation should be `WHERE GOVERNMENT LIKE "%Republic%"`.

**`INNER JOIN` vs. `LEFT JOIN`.** Suppose the utterance is "*List singer names and number of concerts for each singer.*" and the database contains a table of singers and a table with records $(s, c)$ if singer $s$ performed in concert $c$. The SPIDER annotation is incorrect because it uses `INNER JOIN`, which fails to return singers with count 0.

**Ties for Extremals.** For the utterance "*Who is the youngest person?*", the SPIDER annotation is `SELECT NAME FROM PEOPLE ORDER BY AGE LIMIT 1`. As APEL discovers, in case of ties, humans prefer a SQL that will return *all* of the people who have the smallest age, rather than just the first one.

**Remark.** Since most of the text-to-SQL models had low performance 3 years ago, Yu et al. (2018) favored short and plausible SQL annotations to make learning easier. These annotation conventions were shared between training and test sets to form a coherent structured prediction task (internal validity). Now that structured prediction is working well enough that the predictions could be used in real-world settings, we should turn to assuring that the SQL annotations actually have the desired effects (external validity). APEL can help in establishing the new gold standard (§6.2).

## 6.7 Qualitative Feedback

We informally interviewed some of our subjects to obtain feedback about APEL. Here is some example feedback that indicates future room for improvement:

- Boredom: Some subjects complained that the annotation process was boring and they would not want to do it a second time. Future work can design better UIs and interactions to make the annotation more engaging.

- Difficulty: While most questions are straightforward, some require onerous computations (e.g., requires adding 170115 and 50456). Even though we set up a time limit for each question, these questions consumed a lot of mental energy. Future work could include an option for the users to skim through all the questions and solve the easier ones first.
- Vagueness: Some utterances were inherently vague. Consider the utterance "*Count the number of friends Kyle has.*" – what to do when there are two students named `Kyle`? Without external affirmation that these questions were indeed vague, some subjects wasted too much time guessing how to interpret them. Future work could elicit confidence ratings, rather than choosing a discrete option.

## 7 Related Work and Future Directions

Our framework can potentially generalize from text-to-SQL to other semantic parsing tasks. The SQL program $s$ can generalize to other types of executable semantic parses, the input database $i$ can generalize to any well-typed input, and the database query result $o = s(i)$ can generalize to the intended effect of $u$ on $i$. Applying APEL to a new task would require 1) a seed semantic parser with high top-$k$ accuracy, and 2) an algorithm to find a simple informative program input $i$. These problems are related to semantic parsing and programming by example, respectively. We now discuss how those two lines of research benefit from APEL and also how they could make APEL more powerful in the future.

**Semantic Parsing.** Semantic parsers have improved significantly over the past decades (Zettlemoyer and Collins, 2007; Scholak et al., 2021a). Recent pretrained models can perform the task without task-specific architectures (Scholak et al., 2021b) or even in a zero/few-shot manner (Shin et al., 2021; Rajkumar et al., 2022).

However, collecting semantic parsing datasets is still challenging since it requires experts. Wang et al. (2015) addresses this by synthetically generating logical forms, using templates to explain them in natural language, and asking crowdworkers to paraphrase them. Still, the paraphrases are usually restricted in linguistic diversity (Larson et al., 2020). Another line of research (Yao et al., 2020; Elgohary et al., 2020; Mo et al., 2022) tries to learn from user interaction based on the surface form

information, e.g., whether a program refers to specific fields. However, their assumption that a non-expert annotator can recognize the true program $s$ via paraphrases is unrealistic when programs have complex semantics. In contrast, APEL allows non-programmers to indirectly label text-to-SQL data via input-output examples.

Applying APEL to other programming languages requires a seed semantic parser with high top-$k$ accuracy. Such a requirement is more likely to be fulfilled in the future, at least for programming languages that are well-represented in language model training data, as language model capability is expected to improve (Ganguli et al., 2022). Additionally, the seed parser can be improved with better prompting techniques (Trummer, 2022; Wei et al., 2022; Wang et al., 2023; Zhou et al., 2023), which are complementary to our contribution and would make APEL stronger.

**Programming by Example.** PBE has been applied to synthesize regular expressions (Gulwani, 2011), tensor manipulation (Shi et al., 2020), data analysis (Bavishi et al., 2019), and visualization (Wang et al., 2021) programs, etc. Some other recent works such as Ye et al. (2020); Baik et al. (2020) also try to combine semantic parsing with PBE. However, both of them require the users to provide the input-output examples, which can be time-consuming to write.

To reduce the users' workload, we provide the input part of the input-output examples, and focus on only those inputs whose outputs will actually help identify the desired concept. This is a case of active learning, which is broadly used in other applications such as learning real-valued functions, sequence classification, and visual question answering (Schulz et al., 2018; Ein-Dor et al., 2020; Karamcheti et al., 2021). In APEL, for each utterance, we maintain a prior over the function (program) space and learn the desired function by querying it on a sequence of carefully chosen inputs. Similar to our work, Pasupat and Liang (2016) asked non-programmers $o$-selection questions by synthesizing table inputs, but they did not optimize for question simplicity and focused on a simpler single-table setting. Concurrent to our work, Lahiri et al. (2022) applied a similar framework to label Python functions; unlike them, we validated APEL with human annotators.

To reduce the effort required from humans, future work can also use large language models to evaluate the program *outputs* (Chen et al., 2023). Prompting large language models to evaluate their own outputs is a widespread idea. Schick et al. (2023) rerank *ad hoc* programs generated by a language model based on whether their outputs increase the probability of observed text. Bai et al. (2022) automatically critiques model-generated outputs and refines them for future training.

To apply APEL to other programming languages, we need efficient methods to synthesize simple and informative program inputs. Such methods already exist for less expressive languages such as regular expressions or restricted SQL (Miao et al., 2019). Appendix M describes an additional case study where we used APEL (with simulated annotators) to label utterances with regular expressions. To extend APEL to more complicated programs, we can potentially use language models to generate test inputs (Schäfer et al., 2023) and train them to search for more informative inputs with self-supervision (Haluptzok et al., 2023).

**Scalable Oversight.** As AI systems become more capable of generating candidate responses, an emerging line of research supervises AI systems by providing preferences over AI-generated candidates rather than providing human demonstrations (Askell et al., 2021; Ouyang et al., 2022). Therefore, to supervise AI to perform more complex tasks, it becomes increasingly important to determine human preferences over model outputs that are expensive to verify (Amodei et al., 2016; Bowman et al., 2022), such as full-book summaries or natural language descriptions of distributional properties (Wu et al., 2021; Zhong et al., 2022). Our work presents a strategy to re-weight complex outputs from an AI system (namely, programs) by asking simple informative questions of annotators who do not have to understand the outputs directly.

# 8    Conclusion

We proposed APEL, enabling non-programmers to indirectly label SQL programs via input-output examples. With advances in semantic parsing and PBE, future work could potentially extend APEL to other applications. We hope non-programmers can label more complex programs in the future, hence decreasing the costs of supervising increasingly capable AI systems that can take complicated actions by generating and executing code.

## Acknowledgements

The first author is funded by NSF-Simons Theorinet Grant (NSF Award #2031985). Our human interaction study was approved by UC Berkeley's Institutional Review Board, and our survey and interface did not collect any personal identifiable information. We thank members of the Berkeley NLP group and Jacob Steinhardt's group, and the anonymous reviewers for their feedback on our paper draft.

## Limitations

APEL is only a preliminary step towards allowing non-programmers to label utterances with programs. We are not close to enabling expert-level labeling for arbitrary programming languages. We only experimented with English utterances, SQL programs, and university students; generalizing this to more natural languages, programming languages, and annotator populations requires more future work.

Our current implementation is limited to text-to-SQL and regular expressions. As mentioned at the end of the last section, applying APEL to other semantic parsing tasks requires effective algorithms to find simple but informative program inputs and a strong seed semantic parser. While we believe that these assumptions are likely to hold in the future, they might not hold currently. We also assumed that we have access to a pool of unlabeled utterances to begin with, while in practice we might not have access to utterances from real users. More future directions are discussed in Appendices B and I.

As APEL only considers the function (i.e., input-output mapping) computed by a program, it is only able to annotate an utterance with a semantic equivalence class of correct programs. Other factors such as efficiency or readability might be needed to choose among the programs in that class.

Even though we have shown that APEL has higher accuracy than the seed parser, we have not empirically validated that the seed parser is improved by fine-tuning on the programs selected by APEL, nor did we study whether such improvements are in fact cheaper to attain with non-programmers than with expert annotators.

Finally, no semantic parser should be used to synthesize SQL queries or other semantic forms for high-stakes scenarios without a careful analysis of errors and the downstream harms that they might cause.

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

# Supplementary Material

## A  Training on APEL Annotations

As APEL is an annotation method, its ultimate goal is to convert unlabeled utterances $u$ into "supervised" training examples $(u, s)$ for a semantic parser. In this appendix, we discuss how these training examples would be constructed and used.

(In the main paper, we used "example" to refer to examples of the form $(i, o)$, used for programming by example (§2). In this appendix, however, we use "example" to refer to examples of the form $(u, s)$, used as training examples for a semantic parser.)

**Training on MAP Annotations**  For each utterance $u$, APEL solicits indirect annotations and produces a posterior probability distribution $p_T$ over programs. The maximum *a posteriori* (MAP) program is defined by $\hat{s} \stackrel{\text{def}}{=} \operatorname{argmax}_s p_T(s)$.

Most simply, we could train a semantic parser on the inferred annotations $(u, \hat{s})$. Specifically, one option is to use these annotations to fine-tune the seed parser $p_0$. However, this may not be practical for a very large pretrained model such as Codex, in which case the annotations could be used to train a dedicated semantic parser from scratch.

**Downstream Evaluation**  In this paper, we evaluated APEL directly by asking whether its inferred annotations $(u, \hat{s})$ are accurate. We measured this by semantic equivalence $[\![\hat{s}]\!] = [\![s^*]\!]$ (see §4.2) on a small test set of pairs $(u, s^*)$ that were annotated by experts (using APEL). One could instead evaluate the actual impact of the inferred annotations on parsing performance, by training a semantic parser (e.g., fine-tuning the seed parser) on a large training set of APEL-labeled utterances, and evaluating that parser on the test set.

**Beyond MAP**  Rather than evaluating only the MAP annotation $\hat{s}$, we could have evaluated APEL's entire distribution $p_T$ using the log-loss, $-\log p_T(s^* \mid u)$, averaged over the test set of $(u, s^*)$ pairs. This is a more sensitive evaluation, though harder to interpret than exact-match accuracy.

But do we care whether the distribution $p_T$ is accurate beyond just its MAP annotation? Yes: one could train a parser $p$ by maximizing its expected log-likelihood

$$\mathbb{E}_{s \sim p_T}[\log p(s \mid u)] \qquad (8)$$

summed over the APEL-annotated utterances $u$. In effect, this treats $p_T$ as giving a set of weighted "soft annotations" for $u$, not just the MAP annotation. It is equivalent to minimizing the Kullback-Leibler divergence $\mathrm{KL}(p_T \mid\mid p)$ (summed over $u$).

**Iterative Retraining**  Once an improved parser has been trained on the inferred annotations—either the MAP annotations or the soft annotations—it can be used as an improved prior $p_0$ in the update rule (1). This can inform APEL in the selection of future questions.

Furthermore, although APEL's past questions can no longer be changed, we can now reinterpret the annotators' responses to those questions, by using equation (1) to recompute an improved posterior $p_T$ from the improved prior $p_0$. The improved soft annotations from these posteriors can be used to retrain the parser again. This procedure can be iterated to convergence to improve the parser.

This iterated procedure is a principled training method. If soft annotations are used for retraining via equation (8) at each iterations, then it is an instance of the Expectation-Maximization (EM) algorithm (Dempster et al., 1977). (If MAP labels are used, it is an instance of the hard or "Viterbi" approximation to EM.) EM is guaranteed to converge to a new semantic parsing model $p_0$ that locally maximizes the conditional log-likelihood (7) of all of the observed data—in our case, the annotators' responses $r_t$ for all $o$-selection questions on all utterances $u$. Thus, it is simply a log-likelihood training method where the observed data are not direct annotations $s$, but indirect annotations $r_t$. It marginalizes over the latent programs $s$.

**Richer Model of Annotator Behavior**  The annotator model $p(r \mid a, s(i))$ can be jointly retrained with the semantic parser at each iteration of EM, to obtain a higher log-likelihood (7). Improving the annotator model in this way should result in more accurate distributions $p_T$ at the next iteration of EM, and thus a better semantic parser.

Specifically, when we retrain the parser via equation (8), we can also retrain the annotator model to maximize the expected log-likelihood

$$\mathbb{E}_{s \sim p_T}\Big[\sum_{t=1}^{T} \log p(r_t \mid a_t, s(i_t))\Big] \qquad (9)$$

summed over the APEL-annotated utterances $u$. This EM procedure will converge to a maximum of the incomplete-data log-likelihood as in §6.4.

We could fit a richer model of annotator behavior than the relatively simple one in §6.4. For example, we could estimate the error rates of individual annotators on different types of questions and program inputs. Intuitively, equation (9) means that we will tend to judge annotators as correct on examples where they agree with the consensus of $p_T$ and the other annotators.

Although equation (1) assumed that $p(r \mid s, u, a, i) = p(r \mid a, s(i))$, a rich model could drop this assumption.[3] For example, the model might allow that the annotator is more likely to make an error when $u$ is difficult or $i$ is complex. As an annotator's errors on a given utterance may be influenced by how they answered previous questions, a rich model could even take the form $p(r_t \mid s, u, a, i_1, r_1, \ldots, i_{t-1}, r_{t-1}, i_t)$, where $i_1, r_1, \ldots, i_{t-1}, r_{t-1}$ are the previous rounds of interaction with annotator $a$.

**Evaluating with Task Loss** In our discussion of evaluation thus far, we have not given the semantic parser any partial credit. That is, given an utterance $u$, we have considered any answer other than the correct program $s$ to be equally wrong.

However, more generally, it may be tolerable to find a program whose outputs are correct—or at least close to correct—for most inputs. Let $\text{loss}(\hat{o} \mid u, i, o^*)$ denote the task-specific loss of predicting output $\hat{o}$ on input $i$ when the correct output is $o^*$. The loss of predicting program $\hat{s}$ when the correct program is $s^*$ is then

$$\mathbb{E}_i[\text{loss}(\hat{s}(i) \mid u, i, s^*(i))] \qquad (10)$$

where the expectation $\mathbb{E}_i$ is taken over some realistic distribution of inputs for utterance $u$. This loss function can be used in supervised evaluation.

**Decoding with Task Loss** Following standard Bayesian decision-theoretic methods, the semantic parser itself can aim to achieve low loss. No change to the training procedure is needed. Suppose we have trained a semantic parsing model $p(s \mid u)$ by EM as described above. We now need to extract decisions from this model. If the semantic parser is required to translate $u$ into a single program $\hat{s}$ that will be used for all inputs, then the best program to choose is the program that minimizes the **Bayes**

---

[3]This means using $p(r_t \mid s, u, a_t, i_t)$ in place of $p(r_t \mid a_t, s(i_t))$ in equations (2), (3), (7) and (9).

**risk**

$$R_p(\hat{s} \mid u) \stackrel{\text{def}}{=} \mathbb{E}_{s \sim p}[\mathbb{E}_i[\text{loss}(\hat{s}(i) \mid u, i, s(i))]] \qquad (11)$$

This $\hat{s}$ is not necessarily the MAP program. Once $\hat{s}$ is predicted, the output on any given input $i$ is $\hat{o} = \hat{s}(i)$.

In some applications, however, it may not be necessary to choose a single program to use for all inputs. Then the best output $\hat{o}$ to return for input $i$ is the output that minimizes the Bayes risk

$$R_p(\hat{o} \mid u, i) \stackrel{\text{def}}{=} \mathbb{E}_{s \sim p(\cdot \mid u)}[\text{loss}(\hat{o} \mid u, i, s(i))] \quad (12)$$

In other words, $\hat{o}$ is now predicted from $i$ by a consensus of multiple programs.

# B Extensions to APEL Annotation

The following natural extensions could be explored in future work.

**Selecting Questions with Task Loss** A model of task loss as in Appendix A can be used to improve the selection of the next question $i_t$. Instead of maximizing the expected reduction in entropy (equation (4)), we can maximize the expected reduction in the Bayes risk (11).

When $p$ is any distribution over programs $s$ for utterance $u$, define the **minimum Bayes risk** by

$$\text{MBR}(p) \stackrel{\text{def}}{=} \min_{\hat{s}} R_p(\hat{s} \mid u) \qquad (13)$$

At the start of round $t$, we can already achieve a Bayes risk of $\text{MBR}(p_{t-1})$. Once we have the annotator's response $r_t$ to question $i_t$, we will be able to update $p_{t-1}$ to $p_t$ and achieve a Bayes risk of $\text{MBR}(p_t)$, where $p_t$ depends on $i_t$ and $r_t$ via the update (2). Thus, the value of information from asking question $i_t$ is

$$\text{MBR}(p_{t-1}) - \mathbb{E}_{r_t \sim p_{t-1}}[\text{MBR}(p_t)] \qquad (14)$$

This is essentially the same as the information gain (4), but with task loss replacing log-loss. It is again guaranteed to be $\geq 0$.

**Richer Model of Annotator Effort** Our objective (5) tries to present the annotator $a_t$ with a "simple" input database $i_t$, as measured by the number of records $|i_t|$. However, $|i_t|$ is only a crude measure of the time or effort that the annotator must expend to answer the multiple-choice question derived from $i_t$. For example, annotators typically

find it more difficult to perform arithmetic or to reason across multiple tables within $i_t$.

To predict the annotator's effort on the input database $i_t$, we could attempt to simulate the annotator's mental execution of the true program $s$ on $i_t$.[4] As we do not know the true $s$, we would need to take the expectation of this effort over $s \sim p_{t-1}$. Mental operations such as adding 3-digit numbers or visually scanning the tables for a specific string could be then given appropriately high costs in the effort model.

In addition, perhaps a question that generates more multiple-choice options requires more effort. To capture this, the effort model could assign a cost not only to computing $s(i_t)$ but also to finding that answer in the list of options. More generally, the model could attempt to capture mental strategies that do not fully compute $s(i_t)$ but only do enough work to eliminate most of the options.

Finally, a question generally requires less human cognitive effort if it is related to a preceding question. Querying input database $i$ with utterance $u$ is easier for an annotator who has just queried the same $i$ with a different $u'$, or queried a different $i'$ with the same $u$.

How would we train the parameters of the effort model? We observe how quickly the annotator answered each $i_t$ with $r_t$. Taking this time as a proxy for effort, we can train the effort model to predict it from the true program $s$ (which the annotator is presumed to know) and $i_t$. If we do not know $s$, we can impute it from the observed responses—that is, evaluate the effort model's log-likelihood in expectation under $s \sim p_T$, analogously to equations (8) and (9). In short, the effort model could be trained by EM, jointly with the other models.

**Choosing Who Should Annotate What**   The selection objective (5) manages the tradeoff between annotator effort and information gain by imposing a hard constraint on the annotator effort per $o$-selection question. We could improve this crude selection objective—especially given good models of annotator behavior and annotator effort—by selecting a question $i$ that achieves high "bang for the buck." This score is given by the *ratio* of value of information (equation (4) or (14)) to the expected effort of acquiring that information.

Both the numerator and denominator of this ratio depend on the specific annotator $a$ who is an-

swering $i$, if the models have annotator-specific parameters. They also depend on the utterance $u$. Thus, the score is actually a property of the triple $(u, i, a)$.

At each step of APEL annotation, we would select a high-scoring triple—one in which we expect annotator $a$ to usefully evaluate utterance $u$'s desired behavior on input database $i$ with low effort. The annotator's response $r$ then influences the scores of other triples involving $u$. We would stop annotation when no sufficiently high-scoring triple could be found.

In the same way, Bachrach et al. (2012) and Whitehill et al. (2009) model each individual annotator's capability and each question's difficulty, learning these parameters through agreement information. Yan et al. (2011) explore these ideas for active learning, similarly routing useful questions to competent annotators. Our experiment empirically finds that each annotator's ability to answer questions and the difficulty of each domain varies wildly (Figure 10); therefore, future work is likely to benefit from actively selecting who to annotate which utterances.

**Non-Myopic Selection Policy**   The procedure described just above will switch freely among utterances and inputs, if it always selects the highest-scoring triple. However, recall that if a followup question on the same utterance or input is eventually to be asked at all, asking it immediately will incur less effort from the annotator. Thus, a good heuristic is to prioritize followup questions over non-followup questions, provided that they score highly enough that it is likely that they will eventually be asked.

More generally: Greedily choosing the highest-scoring question at each step is common in interactive protocols for information acquisition, such as adaptive quadrature or Bayesian optimization (Schulz et al., 2018). However, this greedy procedure is in general suboptimal. One could do better at selecting the next question by planning ahead to subsequent questions (Chen et al., 2015), though at a higher computational cost.

Again, we leave all these refinements to future work.

## C   Other Synthesis Constraints

This appendix gives further details about our method of database construction (§3). Overall, we

---

[4]The same simulation could be also used to help model errors in the annotator's response $r_t$ (Appendix A).

follow the recipe of Zhong et al. (2020) to generate large informative databases that conform to a given schema $c$. We draw upon the existing sample database with this schema (provided by the SPIDER dataset in our experiments) to obtain plausible cell values. Following Zhong et al., we first synthesize cell values for all the "parent" columns (i.e., the columns that are being referenced by a child column in a foreign key relation), and then populate child columns with random elements from the parent columns.

**Naturalness of $i$**  As shown in Figure 6a, unrestricted random cell values can confuse annotators who are unfamiliar with databases. Therefore, rather than synthesizing completely random values, we now always copy individual cell values from the sample databases (§4.1), optionally with minor perturbations such as $\pm 1$ for integer values.

A database record might also be confusing even if its individual cell values are not. For example, the annotator can be confused by counterfactual information where the U.S. is in Asia as shown in Figure (b). Therefore, we prefer to initialize $i^0$ with the existing database. The annotator can also be confused by uncommon patterns where two people have the same name but different IDs; therefore, if the existing sample database has unique values in a column, we prefer to enforce that $i$ also has unique values in that column.

**Non-vacuous Execution**  Extremely small $i$ frequently leads to undefined denotations. For example, the maximum of zero elements is a special NULL value, meaning "undefined"; this confuses annotators without a computer science background (Figure 6d). Therefore, we always add a small probability mass ($\frac{1}{16}$) of RETURN NULL to the distribution $p$ and normalize the probabilities of the other candidates proportionally, in order to incentivize our algorithm to produce $i$ such that other SQL candidates will return non-NULL values.

Even if the returned value is well-defined, small $i$ can lead to confusion if some operators are not needed to answer the question. For example, in Figure 6e, asking the maximum over one element might appear confusing, as we do not need the max operator to obtain a correct denotation. Therefore, we always add into $p'$ a small probability mass of "neighbor queries" (Zhong et al., 2020) obtained by dropping aggregation operators and WHERE clauses from SQL candidates in $p'$. This incentivizes our al-

gorithm to produce $i$ such that the SQL candidates will meaningfully use their operators.

**Managing Tradeoffs between two Criteria**  All the above tweaks make a tradeoff between the informative and the simplicity criteria in some way: we impose restrictions on $i$ or modify $p'$ to decrease the annotator effort while sacrificing information gain we can potentially achieve. How do we decide when to apply certain tweaks?

In our paper, we always add small probabilities of neighbor queries and RETURN NULL to $p'$ and use cell values from the existing database. We then consider 3 types of tweaks that we apply if possible: 1) $i_0$ satisfies the uniqueness constraint, 2) $i_0$ is initialized with an existing database, and 3) $|i| \leq 15$ rather than 30. We apply these tweaks if they do not prevent us from succeeding. We define in total $2^3 = 8$ different "configurations" $0 \leq c < 8$, each of which specifies what subset of tweaks to apply to the algorithm described in §3. For example, $c = 6 = B110$ means we apply tweaks 1) and 2). We enumerate from $c = 7$ to 0 until the algorithm from §3 returns a program input with $\text{IG}(i) \neq 0$. In other words, we start by applying all the tweaks and drop the tweaks gradually until we obtain a $i$ with positive expected information gain.

# D  Fixing SPIDER Databases

We found several issues with the SPIDER databases and fixed them as follows:

- Some SPIDER databases do not conform to the foreign key constraint, i.e. some of the child columns contain values not in the parent columns they are referring to. We enforce the foreign key constraint by dropping the illegal records.

- In some domains, we identify missing foreign key constraints and add them.

- The voter_1 domain does not contain an appropriate foreign key design. Since fixing it would require an effort of re-annotating all 15 associated SQLs, we chose to exclude this domain from our evaluation.

- Some Date typed columns are string-valued and use English words to represent values, e.g. nov1,2021. As a result, dec1,2021, which is chronologically later, will be considered smaller alphabetically. We fix this

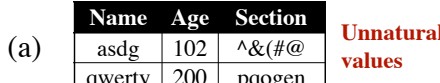

(a) 

| Name | Age | Section |
|------|-----|---------|
| asdg | 102 | ^&(#@ |
| qwerty | 200 | pqogen |

**Unnatural values**

(b)

| Country | Continent |
|---------|-----------|
| U.S. | Asia |
| Canada | Africa |

**Counterfactual content**

(c)

| ID | Name | Age |
|----|------|-----|
| 1 | Eren | 26 |
| 2 | Eren | 23 |

**Uncommon pattern**

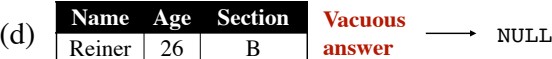

**Utterance**: *How old is the youngest person from section A?*

(d)

| Name | Age | Section |
|------|-----|---------|
| Reiner | 26 | B |

**Vacuous answer** ⟶ `NULL`

(e)

| Name | Age | Section |
|------|-----|---------|
| Eren | 26 | A |

**Vacuous operator** ⟶ `26`

Figure 6: Examples of unnatural databases (top) and vacuous execution (bottom), which motivates several tweaks in Appendix C. In (a) the individual cell values are unnatural. In (b) the records contradict world knowledge. In (c) the database contains two persons with the same name, which is atypical (but possible). In (d) the denotation of the utterance is undefined, since we cannot take the maximum of zero elements. In (e) we do not need the max operator to obtain the correct denotation, since there is only one person in section A.

by canonicalizing date representations into a `yyyy-mm-dd` format.

We accordingly update the suite of test cases from Zhong et al. (2020) that we use to check whether two SQL forms are equivalent (see §4.2), so that they conform to the new database schema.

## E  Generating SQL Candidates

### E.1  Prompting Codex

As sketched in §4.2, we obtain SQL program candidates through few-shot prompting, where the database schema is followed by 4 or 8 (with 50% probability) pairs of natural language utterances with their corresponding SQL queries from the SPIDER development set from the subset of utterance-SQL pairs associated with the same database schema. To select each in-context example, with probability 50% we choose the examples most similar to the utterance $u$ that we want to annotate based on TF-IDF similarity, and with probability 50% we choose a random example that has not yet

| $k$ | easy | medium | hard | extra | all |
|-----|------|--------|------|-------|-----|
| 1 | 0.87 | 0.80 | 0.56 | 0.45 | 0.72 |
| 2 | 0.94 | 0.89 | 0.74 | 0.63 | 0.84 |
| 4 | 0.96 | 0.93 | 0.87 | 0.70 | 0.89 |
| 8 | 0.97 | 0.95 | 0.95 | 0.78 | 0.92 |
| 16 | 0.98 | 0.96 | 0.98 | 0.81 | 0.94 |
| 32 | 0.98 | 0.96 | 0.98 | 0.85 | 0.95 |

Table 3: The top-$k$ accuracy for the SQL candidates generated by Codex on SPIDER (Yu et al., 2018), calculated on each split.

been selected. Finally we append the natural language utterance $u$ to be annotated, and ask Codex to continue generating text after this prompt, which generates a candidate SQL program corresponding to $u$. An example prompt can be seen in Figure 7. As mentioned in §4.2, we sampled 200 different prompts, which varied in their selected examples, and for each prompt we sampled 20 candidates from Codex with temperature=1.0 and top_p=0.95.

### E.2  Top-$k$ Accuracy

As mentioned in §4.2, we defined $p_0$ as a distribution over the top-$k$ approximate equivalence classes of the 4000 sampled programs, where $k = 16$. Table 3 reports the candidate ceilings (§4.4) for various other values of $k$, and Figure 8 graphs these as top-$k$ accuracy curves.

## F  Errors in Non-Programmer Responses

**Ambiguous Utterances.** Consider the utterance "*What are the names of properties that are either houses or apartments with more than 1 room?*" Should it be parsed as "*(house) or (apartment and room > 1)*", or "*(house or apartment) and room > 1*"? Another example: "*Count the number of friends Kyle has.*" What to do when there are two students named `Kyle`?

**Heavy Computation.** It is hard for humans to do arithmetic mentally, e.g., find the average of eight 9-digit values. To avoid demanding such computations, APEL should improve the annotator effort model $|i|$ beyond counting the number of records (Appendix B).

**Database Constraints with Common Sense.** Database schemas sometimes omit common-sense constraints. For example, according to common sense, "BIRTHDAY + AGE" should always yield the current year, so sorting by BIRTHDAY ascendingly

```
CREATE TABLE Highschooler(
        ID int primary key,
        name text,
        grade int)
CREATE TABLE Friend(
        student_id int,
        friend_id int,
        primary key (student_id,friend_id),
        foreign key(student_id) references Highschooler(ID) ON DELETE CASCADE,
        foreign key (friend_id) references Highschooler(ID) ON DELETE CASCADE
)
[Other table schema omitted]

Write a query that answers "Count the number of high schoolers."
SELECT count(*) FROM Highschooler

Write a query that answers "What are the names of high schoolers who have 3 or more
friends?"
SELECT T2.name FROM Friend AS T1 JOIN Highschooler AS T2 ON T1.student_id  =  T2.id
GROUP BY T1.student_id HAVING count(*)  >=  3

[6 More examples omitted]

Write a query that answers "Find the average grade of all students who have some
friends."
___________[Models' Completion] ______
```

Figure 7: An example prompt we use for the Codex API. We obtain SQL program candidates through 4/8-shot prompting, where the database schema (orange) is followed by 4/8 pairs of natural language utterance and their corresponding SQL queries from the SPIDER development set, randomly sampled from the subset of queries associated with the same database schema. Finally we concatenate the target natural language utterance $u$ to be annotated, and ask Codex to complete the prompt, which results in a candidate SQL program corresponding to $u$.

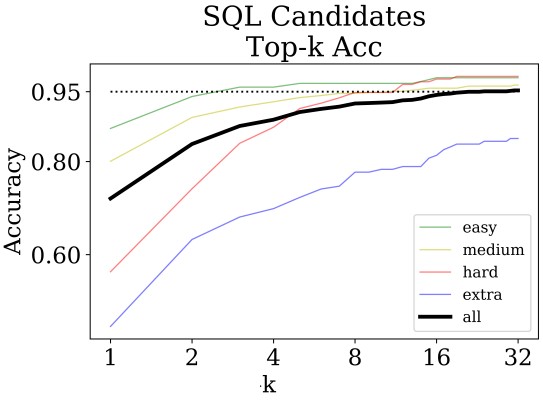

Figure 8: The top-$k$ accuracy for the candidate SQL programs generated by Codex, after filtering and merging candidates. On each difficulty split we plot the curve of top-$k$ accuracy (y-axis) and $k$ (x-axis, log-scaled). The numbers can be seen in Appendix Table 3.

is equivalent to sorting by AGE descendingly. However, APEL is able to find a database where these two strategies return different outputs.[5] Such a database is unnatural and confuses humans. A possible solution would be to induce such constraints from the sample database and/or the column names in the schema.

## G    Computation

We did not compute the runtime in a controlled environment, so the statistics in this section are our best estimate.

Finding a single informative small database can take up to several minutes. The simulation evaluation on the evaluation split in §5 (524 utterances) takes around 240 CPU hours in total.

For the human evaluation in §6 (240 utterances), we must pre-compute the databases for each utterance, in order to support real-time interaction. Since we may ask the annotator up to 3 questions about the utterance, the choice of database

---

[5]Even though APEL derives its databases from the original SPIDER sample database, that sample database contains records that do not conform to this unstated constraint.

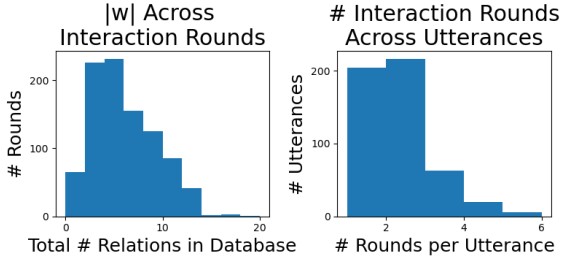

Figure 9: The interaction statistics using the SPIDER annotation to simulate an ideal annotator. **Left**: the distribution of the size of the database $c$ across each round of interaction. **Right**: the distribution of the number of rounds across difference utterance.

|  | easy | med | hard | extra | all |
|---|---|---|---|---|---|
| Candidate | 0.98 | 0.96 | 0.98 | 0.81 | 0.94 |
| Codex | 0.87 | 0.80 | 0.56 | 0.45 | 0.72 |
| OurDB | 0.93 | 0.95 | 0.97 | 0.75 | 0.91 |
| OrigDB | 0.98 | 0.90 | 0.83 | 0.66 | 0.86 |

Table 4: The accuracy ceilings on each difficulty split. "med" stands for medium difficulty. Candidate means the candidate ceiling. OurDB means the accuracy ceiling achieved by querying an oracle annotator with at most 3 small informative databases we synthesize. OrigDB means the accuracy ceiling by querying with the databases released by the SPIDER dataset.

$i_t$ is conditioned on 0–2 previous questions and responses $(i_1, r_1, \ldots, i_{t-1}, r_{t-1})$. We pre-compute the database $i_t$ for each of these possible histories that we may encounter.[6] This takes around 100 CPU hours in total (for the 240 utterances).

## H  Simulated Interaction Statistics

The breakdown statistics of candidate and interaction ceiling (see §4.4) can be seen in Table 4, and the distribution database sizes and number of rounds of interaction can be seen in Figure 9.

**Robustness Towards Hyper-Parameter Choices.** We vary the hyper-parameters in §3 to test its robustness. Changing 5% to 20%, or decreasing the number of random re-runs, all leads to the same

---

[6]Except that we set a timeout of 40 minutes per utterance. Of the 240 utterances, 7 utterances timed out before computing all of the databases in the response tree. (These primarily came from one domain where SPIDER's sample database was very large.) If during the interaction with an annotator, we needed a database $i_t$ that we had not precomputed, we aborted the interaction early (we considered this as a failure to find an appropriate database, in the terms of §2.1). However, this rarely happened, since at each node of the response tree, we considered the most probable responses first.

performance of 91%, and the average number of interaction rounds fluctuates by at most 0.05. Additionally, even after increasing the maximal allowed database size $R$ from 15 to 30, we still obtain an average size of 10, since we prefer smaller databases under the same information gain.

## I  Human Annotation

We provide breakdown statistics on the annotation accuracy of different sources based on difficulty in Table 5.

More analysis on the annotator behavior model can be found in Figure 10.

## J  Interface

See Figure 11 for a detailed screenshot of our interface. We implemented the front-end of our interface with Angular and the back-end was built with flask and Redis. Users are presented with a sequence of 40 distinct questions, and each question may have multiple rounds. For each round, the user is given a 4 minute time-limit before the interface automatically transitions to the next question. Before being asked questions on a new database, the user is presented with a page displaying all the tables in the database alongside descriptions we wrote for each table (see Figure Figure 12 for an example screenshot). When answering questions, the user is given a link back to this page for reference.

The user can either select one of the multiple choice questions presented or select "No Answer is Correct", and depending on their selection the user is presented with a differing set of followup questions. Regardless of their selection, we always ask the user two optional followups: "if you think the question is ambiguous, tell us why." and "if the question looks confusing, tell us why." In addition to these optional questions, we sometimes ask required followup questions. Specifically, if the user is on their final round and selects an answer which does not agree with the SPIDER annotation, we ask them why they did not select the correct answer according to spider. Or if the user selects "No Answer is Correct", we ask "What is the answer you have in mind and why?" We use the users' answers to these followups to collect information on the users' reasoning in answering questions and to determine issues with the SPIDER dataset.

We implemented a number of features in our interface to minimize the annotator effort. One of the largest challenges in this task is answering ques-

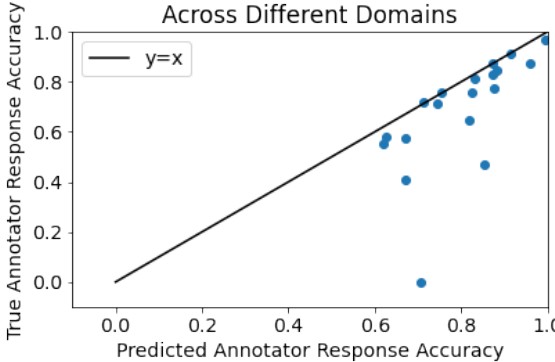 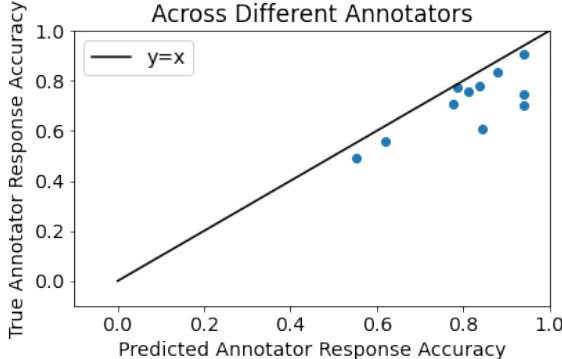

Figure 10: For each domain (left) and annotator (right), we use our annotator behavior model to predict the annotation accuracy ($x$-value) without using the gold annotation and compare it with the true annotation accuracy ($y$-value) using the gold annotation. The outlier at the bottom of the left figure corresponds to a domain that only contains six utterances, where 4 were used for training and 2 used for held-out evaluation.

|  | easy | medium | hard | extra | all |
|---|---|---|---|---|---|
| Candidate Ceiling | 0.91 | 0.91 | 0.92 | 0.69 | 0.88 |
| SPIDER | 0.93 | 0.78 | 0.63 | 0.50 | 0.75 |
| Codex | 0.78 | 0.65 | 0.43 | 0.36 | 0.59 |
| APEL $^r$ | 0.75 | 0.71 | 0.65 | 0.49 | 0.67 |
| APEL $^m$ | 0.83 | 0.80 | 0.73 | 0.47 | 0.75 |

Table 5: 1-best accuracy of various SQL prediction methods, broken down by the difficulty level of the utterance (as categorized by SPIDER). Codex returns the most probable SQL according to $p_0$. APEL $^r$ does the same, after eliminating SQLs that are inconsistent with the responses of a single randomly chosen annotator. APEL $^m$ is our full method, which returns the SQL with the highest posterior probability after we fit our model of annotator error.

tions across several foreign keys. We implement two distinct mechanisms to make this easier for users. First, we highlight all table values or foreign keys matching the value the mouse is currently hovering over. Second, we give the user the option to merge all foreign keys into a single table by pressing a "merge" button. We allow the users to choose when to merge because there is a trade-off; while merged mode can make reasoning about foreign keys easier, it also can significantly increase the width of the tables visible to the user.

Sometimes there are tables presented to the user that are not necessary for answering the question, so we give users the option to collapse tables to simplify their display.

## K    Video Transcript

**Page 1**    In this task, you will be asked to answer questions from several tables.

**Page 2**    Here is the overall idea. You will be given a question on the top of the page, several tables on the left of the page, and you need to choose one of the options on the right, that corresponds to the correct answer. In this question, you are

asked to "Show name, country, age for all singers ordered by age from the oldest to the youngest." Therefore, we expect the correct option to list the information about Joe Sharp first, since he is older. We look at the options and B is correct. Notice that A is wrong because it does not list the information for all singers, and C is wrong because it lists the singers from the youngest to the oldest.

After you submit the answer, our system will ask you whether there is anything that appears ambiguous or confusing. We don't need it for this question now.

**Page 3**    Let's go through some more examples.

**Page 4**    In this question you are asked "How many singers do we have?" This is a tricky question. First notice that the tables have changed from before, so you need to re-read the table. Secondly, there are actually two singers, but they have the same name. You should consider them to be two different persons with the same name but different SSN, and hence choose B.

There is a time limit shown at the top of the page, and after 4 minutes the system will move on to the

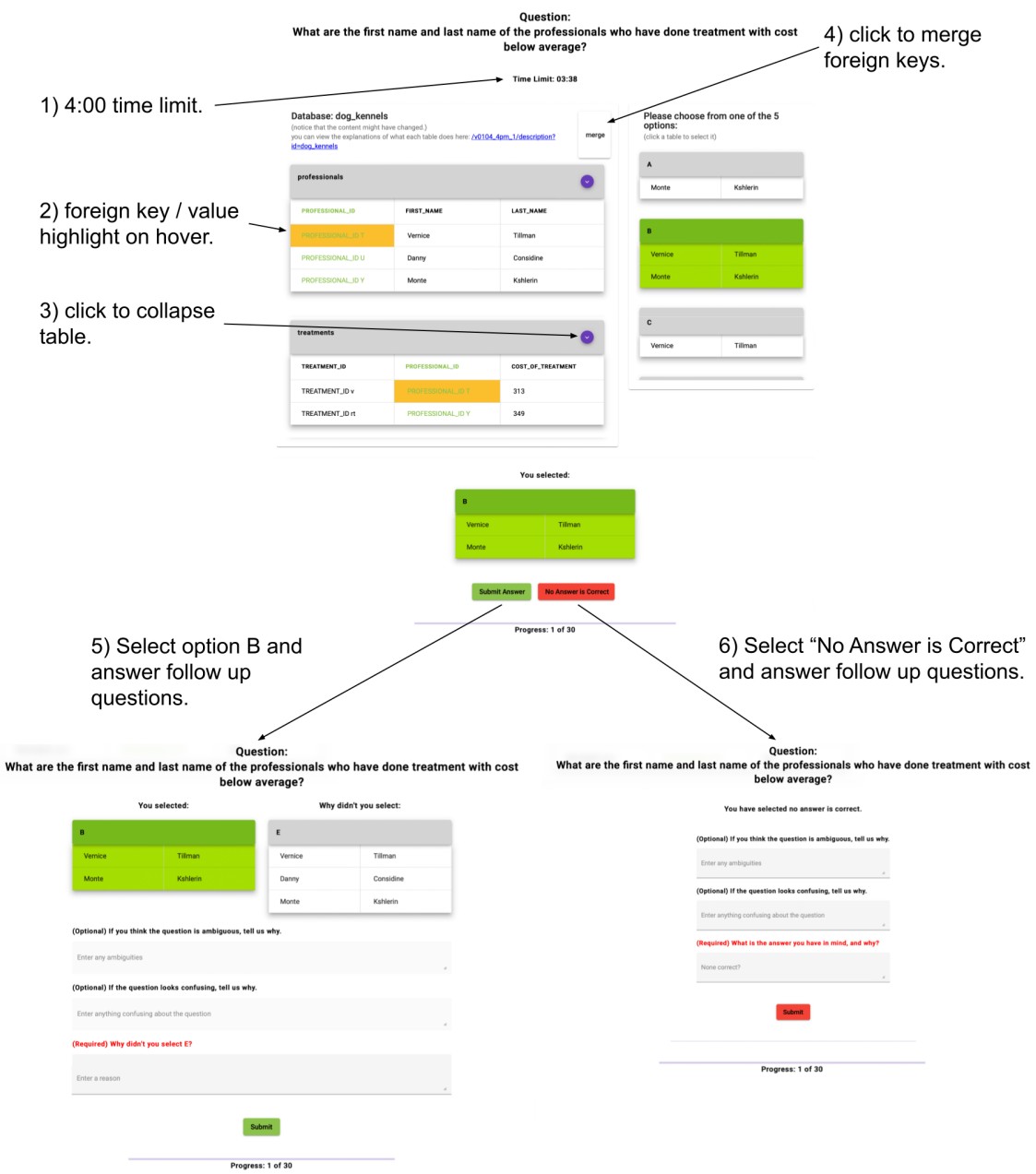

Figure 11: A detailed screenshot of our interface, and the logical flow of follow up questions.

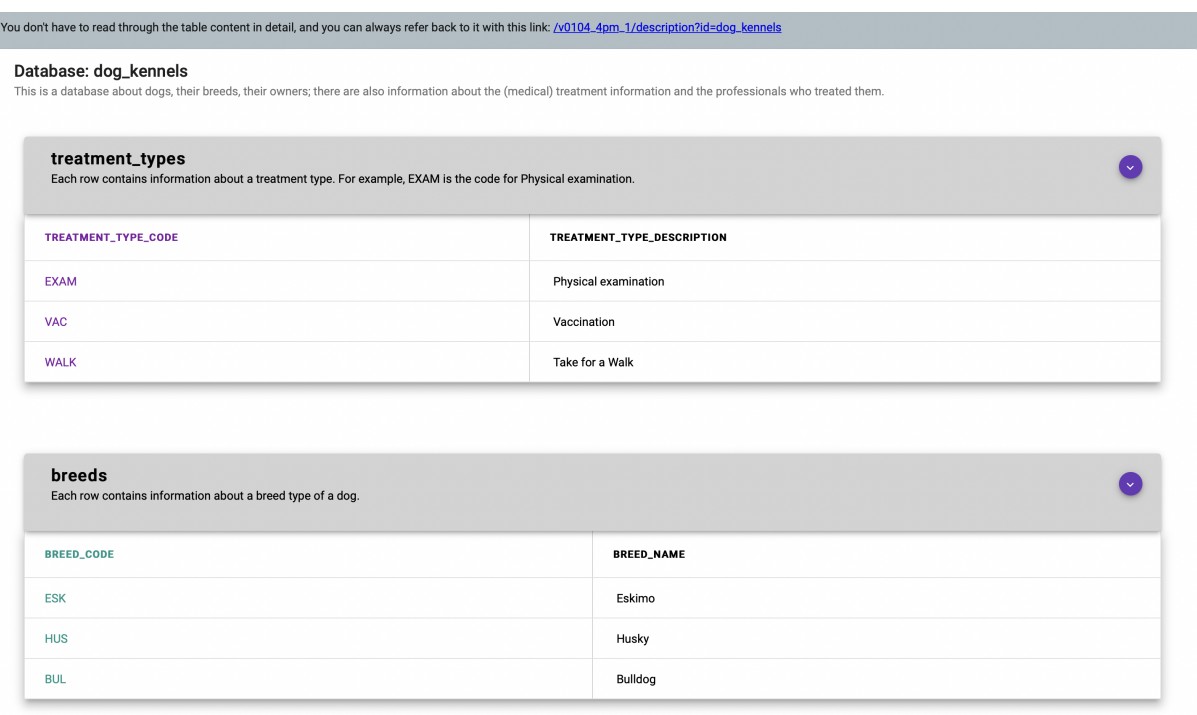

Figure 12: An example database description page presented to users before they start answering questions for that database.

next question.

**Page 5**  Takeaways:

- Names are different from IDs. Two different people can have the same name.

- There is a time limit of 4 minutes for each question.

**Page 6**  In this question you are asked to find the song names of the singers above the average age. The average age is the mean of these 4 numbers, which is 34.5. The singers John and Rose have age above 34.5, so we can find their songs, which are sun and gentle man, which is D. Use a calculator if you need to!

Also, notice that there are other tables, but they are not relevant to the question. Feel free to ignore them. You can also choose to collapse them if that makes it easier, and you can click the button again to view it.

**Page 7**  Takeaways:

- Use a calculator if you need to.

- Not every table is needed.

**Page 8**  Here's the same question and the same table. Let's say somehow Sun and Gentleman is not one of the options, and then you should report that no answer is correct. Then we will ask you why you think no answer is correct. For example, you can write "gentleman and sun is correct. the average age is 34.5, John and Rose are above this age and have song gentleman and Sun".

The system asks us why we didn't choose A, we can answer "sun is by Rose, who is older than 34.5". Please tell us enough information so that we can know why your choice is correct - for example if you just say "sun is also a correct answer", it only describes the difference between the two options rather than explaining why it is correct. Giving us more information can help you win more bonus.

**Page 9**  Takeaways:

- Choose no option is correct and tell us why when you think no options are correct

- Tell us why you didn't choose an answer when we ask you to do so.

**Page 10**  The question is "What are the full names of all players, sorted by birth date?" First, notice that there are a lot of answers in this case, and you need to scroll down to read through all of them.

Secondly, there are a lot of ambiguities: for example, the question didn't mention whether we should sort from youngest to oldest, or the reverse; secondly, the question does not mention whether the first and last name should be in the same column. For these reasons, A, B are both correct. C, D are wrong because the question does not ask for birthday information; F is wrong because it only lists one player and G is wrong for including birthday information. Then we can write in the response: "ABE are all correct; not sure if we should sort them from the oldest to youngest or reverse; also not sure whether to put the first and last name into the same column." But still, make your best guess, let's say, A.

Then we click submit, and the system asks us why we didn't choose C. We explain that "the question does not ask us for the birthday and it contains redundant information".

**Page 11** Takeaways:

- There can be a lot of options. Make sure to read through every of them

- When the question is ambiguous and multiple answers are plausible, tell us why it is ambiguous and what are the plausible answers. But still, first make your best guess and submit.

**Page 12** The question is "Give the names of countries that are in Europe and have a population equal to 80000." In this fictitious table, Brazil is in Europe and has a population of 80,000. Therefore, the correct answer is A, even though we know that Brazil is in fact in South America. However, it still cannot stop us from answering the question based on the table. Finally, there are many more countries in the world, beyond these three countries in the table, but we should pretend that there are only three countries in the world here.

**Page 13** Takeaways:

- Try accepting the information from this table as much as possible and focus on the part useful for answering the question.

- If something is not present in the tables, pretend that it does not exist.

**Page 14** Here are some more difficult tables. This is a database that contains information about battles and death. The overall description of the databases can be seen at the top of the page, which says: This

database contains information about battles, death events, and ships. And then each table has its own description as well. For example, in the ship table, each row contains information about a ship, the 4th row means the ship D was lost in battle with ID 4, and you can look up information about battle 4 in the battle table. To make it convenient for you, whenever you move your cursor to a value, all the same values will be highlighted. Here we notice that according to the 5th row, Ship E was also lost in battle 4.

To view multiple tables at the same time, you can choose to zoom out, like this. Then you can zoom back in, like this. You can typically find this option in the Help panel of your browser. Again, if you think some tables are irrelevant, just collapse them like this.

You don't have to study the tables in detail, since they will probably change for the next question.

**Page 15** Takeaways:

- You don't have to study the table content in great detail, since they will be changing.

- Zoom-in/out if you need to. You can find them in the helper panel of your browser.

**Page 16** This question is "Show names, results and bulgarian commanders of the battles with no ships lost in the 'English Channel".

The question asks for certain battles namely, those that did not lose ships in the English Channel [pause]. Let's start by finding the battles that did lose ships in the English channel [pause]. Only Battle 5 did; it lost ship C there. So the other battles, Battles 0 and 7, lost no ships there. In fact, Battle 0 lost no ships at all, which is why it doesn't show up in the second table. We find the names of Battle 0 and 7, along with their other information. Therefore, the answer is E. One very common mistake people make is that they ignored the word "no", and they chose the battles that lost the ship. Be careful and pay close attention to every word!

Notice that there was originally the death table. We removed it from the display to make it easier for you.

The phrase 'Bulgarian commander' might send you looking for a table that tells you each commander's nationality. But actually, Bulgarian_commander is a column in the battles table. Presumably this table lists battles that Bulgaria fought. Each battle had two sides, and this column

is naming the commander for the Bulgarian side. You don't have to fully understand how the tables are set up, but you should figure out enough to answer the question.

Just to repeat, to make it easier for you to process this information, whenever your cursor moves to an ID or a piece of text, its counterpart in other tables will light up; whenever you click on a text, the counterpart in the answer will also be highlighted.

You can also choose to merge the tables. After you merge the table, there will still be two tables. Each of the rows in the battle table will still contain information about a battle, and each of the rows in the ship table will still contain information about a ship. However, the battle information where the ship is lost is merged into the ship table. Notice that battle 0 will not appear in the ship table, because no ship is lost in the battle, so be careful when you try to interpret the merged table. Click unmerge to recover to the original view.

Finally, if you forgot what each table means, you can always view them here.

**Page 17**  Takeaways:

- Pay close attention to how the question is being asked. They might lead to different options. Many mistakes people make are because they did not read the questions carefully.

- Sometimes we choose not to show you certain tables and columns if we know for sure they are not needed.

- Use the highlight functionality if that helps you to reason across tables.

- Use the merge functionality if you need to. Each table will contain information about the same object/entity, but the information about its related objects will be pooled in.

**Page 18**  The question is "List the name and date of the battle that has lost the ship named 'Lettice' and the ship named 'HMS Atalanta'." Since there is no ship named "HMS atlanta", there is no battle that lost both of these ships. So you should choose A, "no result found".

**Page 19**  Takeaways: Choose no_result_found if no answer satisfies the question.

**Page 20**  To summarize, here are a couple of things you need to remember to answer the questions correctly:

- Pay close attention to how the question is asked; most mistakes are made because of not reading the question carefully.

- Accept the information in the table even if they are changing and might be different from the knowledge you have for the real world

- IDs are different from names

- Some questions might have a lot of options to choose from and you need to read through all of them.

**Page 21**  To make it easier for you to answer the questions:

- Use the highlight and merge operations when you need to

- Use a calculator if you need to

- Zoom out to fit the tables into the screen and prevent scrolling.

- Not all table or column is needed to answer the questions

**Page 22**  For freeform response:

- Reporting ambiguities or tell us why the question is confusing only if you need to

- Explaining why you did not choose another option when we ask you. Giving us more information can you help you win more bonus.

## L   Beyond Text-to-SQL

Our framework can be generalized to other semantic parsing applications more broadly, where

- the SQL program $s$ can be generalized to other types of executable semantic parses, such as tensor manipulation commands, visualization programs (Chen et al., 2021c), or dataflow graphs (Semantic Machines et al., 2020);

- the database schema $c$ can be generalized to include any **c**ontext that affects the mapping of $u$ to $s$, e.g., the conversational history preceding $u$, and the input type required by program $s$;

- the input database $i$ can be generalized to any well-typed input if $s$ is a function, or $i$ can be the program state (e.g., a mapping from variable names to their values) if $s$ is a step in a procedural program;

- the database query result $o = s(i)$ can be generalized to the intended effect of $u$ given $i$, which includes not only an answer or a table, but also an output images, side effects such as file updates (e.g., updates to a database or a document), or robotic actions.

Applying APEL to a new type of semantic parsing application, rather than utterance-to-SQL, would require the following components:

**A seed semantic parser** that is likely to generate a short list of candidates that contain the correct program. This requirement is not hard to satisfy in many applications, given that large language models achieve often achieve high top-$k$ accuracy on generating simple Python snippets (Chen et al., 2021a), JSON data (Poesia et al., 2022), Lispress (Shin et al., 2021) and SQL programs (Scholak et al., 2021b; Rajkumar et al., 2022) with only a few training examples and are likely to continue improving (Kaplan et al., 2020). For example, we achieved 95% top-32 accuracy on SPIDER without any task-specific engineering beyond few-shot prompting (e.g., specialized architectures (Wang et al., 2020), decoding constraints (Scholak et al., 2021b), etc).

**An algorithm** to find an simple informative program input $i$ that satisfies $c$. Our method in §3 generates random databases by using an existing sample database as a reference and greedily drops rows to optimize the objective in equation (5). Future methods could potentially speed up the optimization process with a learned neural network (Chen et al., 2018) or a constraint solver (Chu et al., 2017).

**A graphical interface** that enables the annotators to easily inspect the input $i$ and choose the correct output $o$, where $i, o$ can be generalized from database tables, strings, or numbers to calendar events (Andreas et al., 2020), voxel structures (Wang et al., 2017), etc. Careful interface design (§6.1) can significantly reduce the effort required from the annotators.

In summary, APEL is a general framework for clarifying the semantics of natural language utterances. It elicits information from humans about how the semantic forms should behave when executed on particular inputs.

In this paper we demonstrated the value of APEL on a text-to-SQL task. Some future work is out-lined in Appendices A–B. It would also be desirable to refine and speed up our heuristics for the challenging problem of finding simple inputs that distinguish among SQL queries. Finally, we look forward to future work that extends APEL to other semantic parsing applications.

# M  Simulation Experiments on Regex

We include an additional experiment on regular expressions to help the readers understand how to apply APEL to semantic parsing tasks other than SQL.

**Dataset.** We used the dataset from Ye et al. (2020), which aims to synthesize a regular expression from a natural language description and a few strings that should be accepted and rejected by the regex. For example, an utterance $u$ could be

> "*This is a list of three comma separated strings, where each string must be formed by the substrings "cz" or "rzq" followed by one to four digits.*"

and the corresponding program $s$ is

```
"concat      (      concat  (
or      (       const(<cf>),
const(<rzq>)),repeatrange(<num>,
1   ,4)),concat(<,>    ,concat
(    concat   (    or    (
const(<cf>),      const(<rzq>)),
repeatrange(<num>,   1,    4)),
concat  (  <,>,  concat  (  or
(const(<cf>),const(<rzq>)),
repeatrange(<num>, 1, 4))))))".
```

The program $s$ maps from any input string $i$ to a boolean output of $1$ or $0$. Thus, an $o$-selection question simply asks whether the string $i$ should be accepted or rejected.

We used the development split to prompt GPT-4 (OpenAI, 2023) to create the seed semantic parser. We tested APEL on the test-inclusion split, where the utterances come from the same annotators who annotated the development split.

**Seed Semantic Parser.** We prompted GPT-4 with few-shot demonstrations to create a seed semantic parser. We sampled 50 demonstrations from the development split to simulate a small "training set" of expert labels. For each utterance $u$ we sampled a candidate program from the seed semantic parser 50 times, where each time we randomly

selected 20 demonstrations from the 50 dev split demonstrations as in-context demonstrations. We then automatically filtered out candidate programs that were syntatically invalid and merged semantically equivalent ones (semantic equivalence of regexps can be tested exactly). Finally, we define $p_0$ to be the empirical distribution of the top-10 equivalence classes. The top-1 accuracy (seed parser) is 64% and the top-10 accuracy (ceiling) is 77%.

**Simple and Informative Program Input.** We wish to choose among several candidate programs (or more precisely, equivalence classes). For each pair of candidate programs $s_1$ and $s_2$, we sample 200 program input strings $i$ such that $s_1(i) \neq s_2(i)$, using the efficient implementation from Ye et al. (2020). We then collect all such input strings across all pairs of candidate programs, find the ones with the highest information gain, and break ties by choosing the shortest one as a proxy for simplicity.

To evaluate whether the above procedure is effective in generating simple and informative program input, we consider a baseline that only generates an input string that would be accepted by the highest-ranking candidate regex, again using the implementation from Ye et al. (2020). Intuitively, this baseline tries to check the highest-ranking candidate program, by determining whether a string that it accepts should in fact be accepted. The response (if assumed to be correct) may eliminate some of the programs.

Our information gain method, by contrast, actively tries to distinguish among the candidates. It hopes to gain up to 1 bit of information about the correct candidate (this is the maximum information that can be derived from the annotator's 1-bit response). An input string has IG $\approx$ 1 bit if the candidates that would accept that string currently have about half of the probability mass. In that case, either response (1 or 0) will eliminate about half of the candidates (cf. Littlestone and Warmuth, 1994).

That said, *any* input string $i$ sampled by our procedure will distinguish between two of the candidate programs, and hence will have positive information gain. The annotator's response $r$ (if assumed to be correct) will eliminate one of those two candidate programs, and perhaps others. Thus, it will take at most 9 rounds of oracle annotation for us to rule out 9 incorrect equivalence classes from our original 10 classes, thus achieving the candidate ceiling (§4.4), regardless of which sampled input string we select at each round.

**Results.** After interacting with a simulated annotator for at most three rounds with an oracle annotator as in §5, the APEL accuracy reaches the ceiling of 77% and outperforms the seed parser. In contrast, the baseline only achieves 68% accuracy after three rounds, thus illustrating the importance of optimizing information gain. Finally, APEL uses input strings with an average length of 5.3, while the baseline's average is 11.6; this suggests that optimizing for simplicity is successful.