# OpenReview forum: "Non-Programmers Can Label Programs Indirectly via Active Examples: A Case Study with Text-to-SQL"
_EMNLP/2023/Conference — EMNLP 2023 Main_

### Official Review · Reviewer_uyi9 · 2023-08-04

**Typos Grammar Style And Presentation Improvements:** This is a well-written manuscript.
**Soundness:** 5

**Excitement:**

4: Strong: This paper deepens the understanding of some phenomenon or lowers the barriers to an existing research direction.

**Missing References:**

N/A

**Paper Topic And Main Contributions:**

This paper presents an approach to annotating natural language descriptions with SQL queries using non-programmers.  The authors suggest that, rather than synthesizing programs from whole cloth, non-programmers can be shown example candidate programs as well as input/output pairs capturing program behavior to determine the correct candidate program to use.   The technique involves generating seed programs from given natural language queries.  The seed programs represent (potentially incorrect or incomplete) queries that are then fed to non-programmers.  In addition, seed input data is produced via fuzzing, and the non-programmer helps determine which seed program is correct by considering the natural language query, input, and corresponding output.

The authors demonstrate the viability of their approach using a human study of 11 non-programmers.  The authors show the protocol used and share training materials.   They report their approach can help label natural language queries with SQL queries as accurately as existing techniques requiring programming expertise.

**Questions For The Authors:**

- Were there any qualitative methods or results used in the human subject study? Structured interviews? Post-surveys? etc.
    - The rebuttal addresses these questions.  As for my comment about low participant pool size, I would reiterate that this is, in general, a small size.  One best paper at NAACL (which is not HCI, incidentally) is not convincing given other studies which much larger participant pools in the HCI, AI, and SE communities.  There is perhaps some subtlety relating to the degree to which each participant is involved.  For example, smaller participant pools might be acceptable for studies with a more ethnographic approach.   Regardless, I don't think this is a dealbreaker for this submission.
- Can you describe in more detail how APEL could generalize to non-SQL settings?  In broad strokes, I can understand how you could generate candidate programs and inputs, but for a non-programmer participant, how would something like APEL show, say, C/C++ programs?  What level of abstraction would be considered? Single function? A few lines of code? A whole compilation unit?
    - The authors clarified scope in their rebuttal.

**Reasons To Accept:**

- Timely and relevant topic -- software engineers may increasingly depend on automation to produce candidate programs (e.g., through Codex / Copilot-like technologies).   Thus, is it important to understand how people can help guide automated program synthesis techniques as described in this paper.
- Well executed evaluation.  The authors show a significant increase in accurately labeling natural language queries with SQL programs with respect to reasonable baselines.
- highly intuitive technique based on reasonable insights about active learning and human-in-the-loop labeling.
- Well-written and well-organized manuscript with a reproducible protocol.

**Reasons To Reject:**

- Very small human subject pool for this type of work.  While the authors report statistical significance, I am a bit skeptical about possible coincidences of performance achieved.
- No qualitative methods used in human subject study.  There is a missed opportunity to learn more from the non-programmer participants about their approach to labeling.
- possible overclaiming about the paper's generalizability to new languages.  In this setting, Text-to-SQL involves a fairly direct path for executing candidate seed programs to generate outputs.  It is less clear how (1) intuitive inputs for non-programmers would be selected in non-SQL settings, and (2) how candidate programs would be executed to provide useful outputs for non-programmers to assess each candidate.  The authors discuss programming by example in the related work, but I am skeptical that this technique could extend to new programming languages without a significant investment of research effort into understanding the best way to present inputs and candidate programs.

**Reproducibility:**

5: Could easily reproduce the results.

**Reviewer Confidence:**

4: Quite sure. I tried to check the important points carefully. It's unlikely, though conceivable, that I missed something that should affect my ratings.

---

> ### Author Rebuttal · Authors · 2023-08-28
>
> > Any qualitative interviews?
>
> Thanks for this very insightful comment. We did not have a formal interview process in our experimental design, and retrospectively it was indeed a missed opportunity.
>
> However, we did talk to many of our subjects after the study, and we can discuss them in the updated version. Some example insights include:
> - Some subjects complained that the annotation process was boring and they would not want to do it a second time. Future work can design better UIs and interactions to make the annotation more engaging.
> - While most questions are straightforward, some examples are particularly hard (e.g., requires adding several 10-digit numbers). Though we set up a time limit for each question, these questions consumed a lot of mental energy. Future work could include an option for the users to skim through all the questions and solve the easier ones first.
> - Some utterances were inherently vague. Without external affirmation that these questions were indeed vague, some subjects wasted too much time trying to guess how to interpret the question. Future work could include an entry for the subjects to indicate their confidence, rather than choosing a discrete option.
>
> > The result is found in a small subject pool. Is this a coincidence?
>
> As acknowledged in your review, we conducted a statistical significance test. We found both a substantial effect size (59%->75%) and a significant p-value of < 1e-3 (Table 2), so it is unlikely to be just a coincidence.
>
> In addition to human evaluation, we corroborated our finding with a larger set of utterances with simulation. Section 5 ran larger scale evaluation on 520 utterances in the development set with simulated human annotators and Appendix M evaluated on 630 utterances for regex expression. In both cases, the APEL accuracy significantly outperforms Codex’s Pass@1, indicating the robustness of our conclusion.
>
> Finally, we would like to note that 11 subjects are not particularly small within the HCI literature; for comparison, the NAACL 2022 best paper [1] only involved 9 subjects.
>
> > How does APEL generalize to non-SQL settings?
>
> Empirically, we provided additional experiments on Regex in Appendix M, and a concurrent work [2] used a similar framework to synthesize python snippets (evaluated with simulated humans). We believe that it is also conceptually straightforward to synthesize function snippets in some other languages, such as pandas, pytorch, numpy, matplotlib: in these languages, the inputs and outputs are intuitive to understand, and prior works have studied programming by example in these domains (L603). We agree with you that it is still challenging to extend APEL to any programming language for any programming context (e.g. an entire compilation unit), but we do not make such an unwarranted claim either (see below).
>
> > Possible Overclaiming about generalizability to other languages?
>
> Our submission does not try to claim that “APEL can enable non-programmers to program for any application”. To prevent potential misinterpretation, our submission has explicitly restricted the scope by putting the phrase “a case study in Text-to-SQL” in the title, emphasizing the limitation at the end of the intro (089), and discussing future work in Section 7, the Limitation section, and Appendix Section I. **If specific sentences seem misleading/overclaiming, feel free to point them out so that we can update them for our next version.**
>
> We absolutely agree that in order to label more complicated languages (e.g., C++) in more complex scenarios (e.g., entire compilation unit), “significant investment of research effort into understanding the best way to present inputs and candidate programs.”, as we have also pointed out at the end of our intro (L089) “APEL is a preliminary step towards enabling non-programmers to label arbitrarily complex programs.” To inform future work, we made our best effort to concretize how APEL should be applied to other programming languages by providing an additional experiment in Regex in Appendix M and discussing other languages in Section 7.
>
> While APEL itself is unlikely to be the ultimate solution, the following contributions from APEL will likely endure:
> - We designed an evaluation protocol by comparing machine-assisted non-expert annotations with expert annotations.
> - We proposed to label indirectly when direct labeling is challenging.
> - We used execution results to help non-programmers supervise semantic parsers,  connecting the paradigm of semantic parsing and programming by example.
> - We optimized the questions for the annotators to maximize information gain while keeping it simple.
>
> We look forward to future work that revises and extends our framework to more complex programming tasks.
>
>
> [1] Automatic Correction of Human Translations
>
> [2]  Interactive Code Generation via Test-Driven User-Intent Formalization
>
> [3] DS-1000: A Natural and Reliable Benchmark for Data Science Code Generation

---

### Official Review · Reviewer_EWXL · 2023-08-04

**Soundness:** 5

**Excitement:**

4: Strong: This paper deepens the understanding of some phenomenon or lowers the barriers to an existing research direction.

**Paper Topic And Main Contributions:**

The authors propose a semantic parsing (program) annotation approach that can be carried out by non-programmers, offering a potentially scalable way to collect difficult annotations for tasks that typically require expert annotation. The key insight of their proposed approach, APEL, is to find a simple synthetic program input database that leads to different outputs on candidate programs being evaluated for a given utterance. Non-programmer annotators can then select the program that leads to the correct answer without reading the program itself. The process is repeated iteratively. In a case study on SPIDER development data, the resulting system reaches the same accuracy (75%) as that of database experts.

The authors formalize the APEL approach to select a candidate program input (i.e. database) as maximizing the expected information gain at each step, i.e. inputs that lead to different results between candidates, given a constraint of containing at most 30 records (to reduce annotator burden). Their approach relies on a high-quality seed parser (in their case, Codex with few-shot prompting). In addition to the case study with human annotators, the authors validated their approach using a simulation-based approach.

**Reasons To Accept:**

* The paper is well-written with key ideas presented up front, claims are not overstated
* An ablation on the quality of database selection (as compared to SPIDER’s original database) is conducted that helps validate the effectiveness of the approach
* Extensive analysis is provided on the case study performed on the SPIDER development data
* The paper provides extremely thorough implementation details
* There is clear discussion of annotator demographics and compensation.
* The limitations are clearly stated – a reasonably good seed parser is needed as well as an efficient procedure to find simple/informative program inputs.

**Reasons To Reject:**

* The comparison between Codex P1 accuracy and APEL (which uses human annotators based on a seed parses from Codex) makes sense if we interpret Codex P1 as a competing semantic parse collection strategy. However, in a practical setting we need to trade off the costs of human annotation through APEL vs. the additional volume data that could be collected by Codex by training semantic parsers with the resulting data and applying it to new domains. This is touched on briefly in the limitations and the Appendix.
* It's unclear how repeatable the process of uncovering subtle errors with APEL is (Section 6.6), as these were uncovered by experts (two authors) – this is a nice result, but what makes this specific to the APEL approach is a bit unclear from the main body of the paper.
* Only one (small) dataset with 240 utterances is investigated (worth noting that the Appendix shows how the approach can be extended to other applications)

**Reproducibility:**

5: Could easily reproduce the results.

**Reviewer Confidence:**

4: Quite sure. I tried to check the important points carefully. It's unlikely, though conceivable, that I missed something that should affect my ratings.

**Typos Grammar Style And Presentation Improvements:**

* 050 - then -> to
* 329 - how often are any of the top 16 candidates correct
* Missing reference (??) in footnote on page 13

---

> ### Author Rebuttal · Authors · 2023-08-28
>
> Thanks for reading our paper comprehensively and appreciating our contribution.
>
> > The errors in Section 6.6 were found by experts. How does it contribute to the presentation of APEL, and how repeatable it is?
>
> Re: why we presented it. Since Table 2 claimed that the previous annotator accuracy is only 75%, it might be useful for the readers to qualitatively understand what annotation errors were present.
>
> Re: how repeatable it is with non-experts. All mistakes presented in Section 6.2 were actually also uncovered by non-experts, since the mistakes in the program became obvious when APEL presents an input-output example. Nevertheless, experts were required to categorize and present them. We will make this clearer in the updated version.
>
> Finally, APEL might be useful not only for non-programmers but also for programmers, as debugging might be easier with simple input-output examples. Future work can more systematically study how APEL could help experts find subtle bugs.
>
> > Only one (small) dataset with 240 utterances is investigated
>
> Our results achieved both a substantial effect size (59%->75%) and a significant p-value of. < 0.001 (Table 2), so we believe our conclusion is sound. We did not have enough resources to collect large-scale high-quality human annotations.
>
> In addition to human evaluation, Section 5 ran larger scale evaluation on 520 utterances in the development set with simulated human annotators and Appendix M evaluated on 630 utterances for regex expression. In both cases, the APEL accuracy significantly outperforms Codex’s Pass@1 with simple inputs, indicating the robustness of our conclusion.
>
> > Tradeoffs between the costs of human annotation in APEL vs. the amount of additional data we can collect.
>
> Thanks for bringing this up! Our paper did not study the exact economic benefit of APEL in a deployment environment. However, it provides an alternative, which collects semantic parsing data with non-experts, who might be cheaper and require less time to train in practice. We will acknowledge this limitation in our updated version, and look forward to future work that uses APEL-related methods for realistic downstream applications.

---

### Official Review · Reviewer_Vovf · 2023-08-06

**Soundness:** 4

**Excitement:**

5: Transformative: This paper is likely to change its subfield or computational linguistics broadly. It should be considered for a best paper award. This paper changes the current understanding of some phenomenon, shows a widely held practice to be erroneous in someway, enables a promising direction of research for a (broad or narrow) topic, or creates an exciting new technique.

**Paper Topic And Main Contributions:**

* This work studies a new scenario of improving semantic parsing with labels from *non-expert* programmers. They use text-to-SQL as an example for the study.

* This work proposes APEL framework which leverages the annotation on *program executions* which does not require the annotators to understand the SQL. It has the following steps: 1. generate the seed programs 2. generate synthetic databases that could maximize the information gain for selecting the candidate programs 3. update the posterior probability of the candidate programs. Repeat 2, 3.

**Questions For The Authors:**

* Please see the second bullet point in the above section
* L297+ "We randomly sampled 200 prompts for u by choosing different (uk , sk )". Does this mean there are 200 in-context examples for each test utterance? If so, why choose such a large number?
* Why feeding the original Spider database in oracle setting (Table 1)?
     * The paper mentioned that "Our method also generates much simpler databases than the sample databases, which contain on average 33,295 records and prevent the human annotators from selecting the correct output effectively" but isn't this the **oracle** setting where the correct response is **always** selected (L350-L352)? How is a human annotator related here?
     * "our database generation method leads to higher accuracy since it optimizes information gain" -- my question is that since the original DBs are way larger, won't they cover more possible data combinations and serves as a superset of the synthetic databases?

**Reasons To Accept:**

* The setting of utilizing labels from non-experts is interesting. To my best knowledge, it is not well studied in the literature as well.
* The idea of asking annotators to select the correct *program execution* is intuitive and neat.
* The proposed APEL framework is well-motivated and mathematically sound

**Reasons To Reject:**

* The experiment lacks comparisons with existing approaches. For instance, existing works such as self-consistency [1] and CodeT [2] use execution consistency to select the correct programs. In APEL case, the synthetic program input $i_t$ can be treated as inputs in each round. How does APEL compare to such methods which do not require annotators in the loop?
* The non-standard split of the SPIDER dataset is not well explained and indicates the limitation of applying APEL to zero-shot settings. As mentioned in L315-L320 "These numbers are not comparable to prior works, which usually evaluate on unseen database domains in a zero-shot manner (harder than our setting) but do not require predicting string literals and DISTINCT keywords, which we need for execution." I am wondering (1) how much information APEL needs from the database, only the schemas or the data as well? (2) wouldn't the execution only need the synthetic databases?

[1] Self-Consistency Improves Chain Of Thought Reasoning In Language Models

[2] Codet: Code Generation With Generated Tests

**Reproducibility:**

3: Could reproduce the results with some difficulty. The settings of parameters are underspecified or subjectively determined; the training/evaluation data are not widely available.

**Reviewer Confidence:**

4: Quite sure. I tried to check the important points carefully. It's unlikely, though conceivable, that I missed something that should affect my ratings.

---

> ### Author Rebuttal · Authors · 2023-08-28
>
> Thanks for appreciating our framework and the problem we proposed!
>
> As you have mentioned, our core contribution is a framework that pieces multiple components together (e.g., semantic parser, database generator) to indirectly elicit program labels. We consider **recent advancements on semantic parsers to be complementary** to our contribution and will improve APEL.
>
> Re: soundness, **our submission does NOT claim that the specific system we built is the current SOTA**. Our user study was run in 2021 right after codex-001 was released, a year earlier than [1, 2]; therefore, the exact system we built is unlikely to be SOTA after two years, as do most other ML systems.
>
> Re: excitement, **our conceptual takeaways will likely have lasting value**, even though our current system is not SOTA. As AI will support humans on more difficult tasks, supervising them is challenging for humans because direct labeling is difficult. Our paper is a first step tackling this general problem.
>
> > “Lacks comparisons with approaches such as self-consistency [1] and CodeT [2]”
>
> [1,2] are semantic parsing methods that are complementary to our contribution. We focused on the following question: given a seed semantic parser, such as [1, 2], how to use human non-experts to continue improving it. Nevertheless, thanks for bringing up [1, 2] and our updated related work will include them.
>
> > Why few-shot learning on SPIDER instead of 0-shot? Does APEL also need database values? Is that a limitation of APEL?
>
> The mathematical framework of APEL **requires neither few-shot learning nor the database values**. The specific system we built in 2021 did require few-shot examples to achieve reasonable top-k accuracy since it was based on Codex-001, and it did require values from the SPIDER dataset to generate realistic-looking data. However, future work can use semantic parsers with stronger 0-shot performance and database values generated by language models [3].
>
> > Clarification: L297+ 200 in-context examples for each test utterance?
>
> We retrieved in-context examples from a pool of ~25 examples (50% of the dev set per database), and created 200 prompts by selecting random selections and permutations of 4 or 8 of them to denoise (E.1). Future developers using APEL can start with a stronger semantic parser that does not require significant prompt-engineering effort.
>
> > Clarification: “We require the model to predict DISTINCT keywords and string literals.” Wouldn’t the execution only need the synthetic databases?
>
> L317 points out that the previous evaluation [4] does not require the SQL to be complete (e.g., predict the string literals). However, our paper does require the SQL to be complete to determine the correct semantics, so we reported a stricter evaluation metric.
>
> > Clarification: Why is the human annotator mentioned (L350-L352) if  the correct response is always selected?
>
> Indeed, Section 5 used a simulated perfect annotator instead of human annotators. L350 was a side comment which emphasizes that such a simulation assumption is unrealistic for humans using the baseline – large databases from SPIDER.
>
> > Clarification: Larger databases always have high information gain?
>
> Consider a Table with two columns (student_name, pet_name) where each row means that student_name owns pet_name. Consider two utterances U1,U2 and databases D1, D2
>
> U1: “Find the set of all students” → SELECT DISTINCT STUDENT_NAME FROM TABLE
>
> U2: “Find the set of all students with at most 1 pet” → SELECT STUDENT_NAME FROM TABLE GROUP BY STUDENT_NAME HAVING COUNT(PET_NAME) <= 1;
>
> (D1)
>
> | student_name | pet_name |
> | ----------- | ----------- |
> | Alice | Dog |
> |Alice | Cat |
>
> (D2)
>
> | student_name | pet_name |
> | ----------- | ----------- |
> | Alice | Dog |
>
> D1 is larger than D2; however, U1 and U2 have the same correct answer on (D1) but not on D2.
>
> > Clarification: Why feed the original Spider database in oracle setting (Table 1)?
>
> Even a perfect annotator might not achieve 100% SQL annotation accuracy, since the generated databases might fail to make two candidates return different results. Table 1 compares the database generated by our method to the SPIDER’s database to show that it is effective to optimize databases for simplicity and information gain.
>
> [1] Self-Consistency Improves Chain Of Thought Reasoning In Language Models
>
> [2] Codet: Code Generation With Generated Tests
>
> [3] Language Models are Realistic Tabular Data Generators
>
> [4] Semantic Evaluation for Text-to-SQL with Distilled Test Suites

---

### Meta-Review · Area_Chair_RDne · 2023-09-18

**Recommendation:** 5

**Metareview:**

This paper presents a method for annotating (and improving) semantic parsing with labels from non-programmers, using text-to-SQL. They enable these non-expert annotations by having annotators judge program execution, rather than programs directly. They introduce the APEL framework, which searches for synthetic inputs that lead to different outputs on candidate programs, and then elicit correctness judgments that can be used to fine-tune parsers. They find in a case study that their method can reach the same annotation accuracy as annotations using experts.

The reviewers are in agreement that the paper is well-written, with an interesting, intuitive, and novel framework and methodology. The reviewers also express that the paper features clear presentation of implementation details, extensive analyses, and sensible validation of the approach. Some questions and concerns were raised, but were largely addressed in discussion. Overall I judge that this is a strong submission and a worthy contribution to the conference.

---

### Decision · Program_Chairs · 2023-10-07

**Decision:**

Accept-Main

**Comment:**

This paper presents a method for annotating (and improving) semantic parsing with labels from non-programmers, using text-to-SQL. They enable these non-expert annotations by having annotators judge program execution, rather than programs directly. They introduce the APEL framework, which searches for synthetic inputs that lead to different outputs on candidate programs, and then elicit correctness judgments that can be used to fine-tune parsers. They find in a case study that their method can reach the same annotation accuracy as annotations using experts.

The reviewers are in agreement that the paper is well-written, with an interesting, intuitive, and novel framework and methodology. The reviewers also express that the paper features clear presentation of implementation details, extensive analyses, and sensible validation of the approach. Some questions and concerns were raised, but were largely addressed in discussion. Overall I judge that this is a strong submission and a worthy contribution to the conference.